# Assessing MAPPs assay as a tool to predict the immunogenicity potential of protein therapeutics

Andrea Di Ianni[1,2] , Tiziana Fraone[2], Piercesare Balestra[2], Kyra Cowan[3], Federico Riccardi Sirtori[2] , Luca Barbero[2]

**MHC-II-associated peptide proteomics (MAPPs) is a mass spectrometry-based (MS) method to identify naturally presented MHC-II-associated peptides that could elicit CD4+T cell activation. MAPPs assay is considered one of the assays that better characterize the safety of biotherapeutics by driving the selection of the best candidates concerning their immunogenicity risk. However, there is little knowledge about the impact of bead material on the recovery of MHC-II MS-eluted ligands in MAPPs assays. Here, we firstly describe a robust MAPPs protocol by implementing streptavidin magnetic beads for the isolation of these peptides instead of commonly used NHS-activated beads. Moreover, we assessed the impact of the cell medium used for cell cultures on the morphology and recovery of the in vitro-generated APCs, and its potential implications in the amount of MHC-II isolated peptides. We also described an example of a MAPPs assay application to investigate drug-induced immunogenicity of two bispecific antibodies and compared them with monospecific trastuzumab IgG1 control. This work highlighted the importance of MAPPs in the preclinical in vitro strategy to mitigate the immunogenicity risk of biotherapeutics.**

## Introduction

Immunogenicity is a term used in the pharmaceutical industry to describe the undesired immune responses to therapeutic proteins (Bhogal, 2010). It has been over three decades since the United States Food and Drug Administration (US FDA) approved the first monoclonal antibody in 1986 (Grilo & Mantalaris, 2019; Lu et al, 2020). Over the years, several health authorities have become more interested in how pharmaceutical companies deal with this issue (Kloks et al, 2015). Despite the attempts in tackling the immunogenicity-related potential of protein therapeutics, like using human or humanized protein sequences, the human immune system can still recognize the biological drug product as "non-self," thus triggering an immune response against it (Janeway, 1989;

Kropshofer & Singer, 2006; De Groot & Scott, 2007; Krishna & Nadler, 2016). The development of an anti-drug antibody (ADA) immune response can impact drug pharmacokinetic (PK) and pharmacodynamic (PD) properties (Büttel et al, 2010, 2011). The consequences of these ADAs can range from minor or nonsignificant clinical effects to severe disorders such as deficiency syndromes, thrombocytopenia, and pure red cell aplasia (Macdougall, 2005; Shin et al, 2011; Deehan et al, 2015). The increased interest of regulatory agencies has brought to the publication of guidelines for evaluating immunogenicity to mitigate the harmful effects on patient health (CHMP, 2017; Immunogenicity Assessment for Therapeutic Protein Products U.S. Department of Health and Human Services Food and Drug Administration Center for Drug Evaluation and Research Center for Biologics Evaluation and Research, 2014; U.S. Food & Drug Administration, 2022). Immunogenicity is thought to be influenced by a combination of factors, including characteristics of the drug itself (e.g., nonhuman sequence, glycosylation [Monzavi-Karbassi et al, 2003; Beck et al, 2008], impurities [Mueller et al, 2009; Verthelyi & Wang, 2010], aggregation [Rombach-Riegraf et al, 2014], host cell proteins [Jawa et al, 2016]), patient-specific factors (e.g., type of disease, genetic factors, concurrent use of immunomodulators [Krieckaert et al, 2010]), and the method and frequency of drug administration (Tovey et al, 2011). Furthermore, the incidence of ADAs and their clinical sequelae can completely differ among the same therapeutic modalities and among specific patient populations. Consequently, it is paramount to have reliable preclinical in vitro predictions of immunogenicity (Joubert et al, 2016). The need for better and more consistent preclinical tools to assess immunogenicity risk in humans is even more crucial because animal species used in toxicological studies have shown poor correlation with human clinical data (Gokemeijer et al, 2017).

The current practice of immunogenicity screening generally begins with an in silico assessment and then proceeds with different in vitro/ex vivo assays as needed (reviewed in Jawa et al [2013] and Di Ianni et al [2023]). None of these methods alone can predict the immunogenic properties of new protein candidates with good accuracy. Nevertheless, the combined use of different orthogonal in silico/in vitro assays represents an asset in the

[1]Molecular Biotechnology Center, Department of Molecular Biotechnology and Health Sciences, University of Turin, Turin, Italy   [2]NBE-DMPK Innovative BioAnalytics, Merck Serono RBM S.p.A., An Affiliate of Merck KGaA, Darmstadt, Germany   [3]New Biological Entities, Drug Metabolism and Pharmacokinetics (NBE-DMPK), Research and Development, Merck KGaA, Darmstadt, Germany

Correspondence: andrea.diianni@unito.it; luca.barbero@merckgroup.com

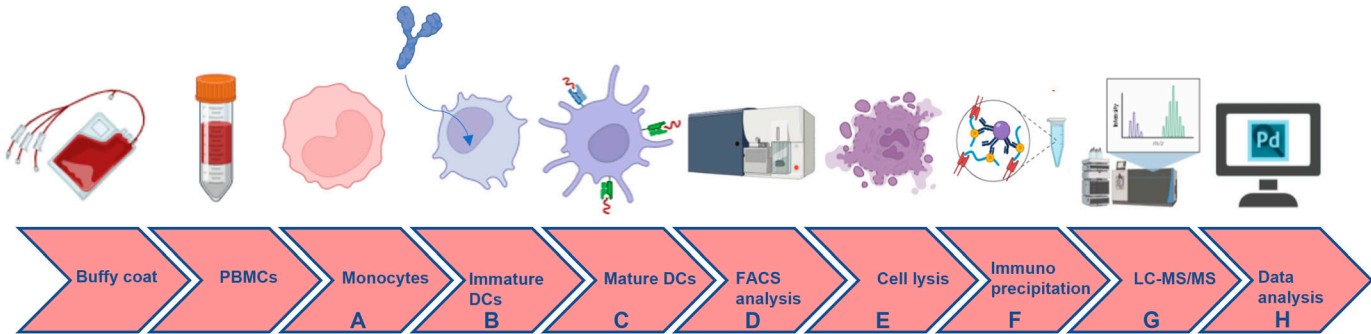

**Figure 1. Typical MHC-II–associated peptide proteomics assay workflow.**
The experimental protocol is composed of a cell-based step and an immunoaffinity-based LC-HRMS step. **(A, B, C)** Isolation of monocytes (CD14[+] cells) from hPBMCs from buffy-coats with anti-CD14 microbeads; (B) differentiation of monocytes into monocyte-derived immature dendritic cells (imMo-DCs) by IL-4 and GM-CSF stimulation and challenging of the achieved imMo-DCs with the antibody of interest; (C) LPS-induced maturation of imMo-DCs to get mature DCs (mMo-DCs). **(D, E, F)** Flow cytometry analysis of dendritic cell surface markers; (E) cell lysis to extract and solubilize the transmembrane peptide-MHC-II complexes; (F) immunoaffinity step via an anti-HLA-DR antibody clone coated onto magnetic beads to recover the peptide-MHC-II complexes, followed by elution of the peptides through an acid-based solution (e.g., trifluoroacetic acid 0.1%). **(G, H)** LC-MS/MS analysis in data-dependent acquisition (DDA) at high resolution to enable the identification of peptides' sequences which have been loaded onto the MHC-II receptors during the assay; (H) data analysis of identified peptides and presentation of results. Reprinted from Di Ianni et al (2023).

concrete understanding of the overall immunogenicity risk of therapeutic agents. In silico tools are commonly used in the early discovery phase to identify potential agretopes (part of the antigen that is processed and loaded onto MHC-II molecules) and drive the candidate's sequence optimization (Peters et al, 2020). Although straightforward and high-throughput, these are likely over-predictive and must be cross-validated using a tailored panel of in vitro assays (Mazor et al, 2015).

These include MHC-II binding assays with synthetic immunogenic peptides (Steere et al, 2006) and cell-based assays also to evaluate the potential for inducing T cell activation/proliferation (Kropshofer & Singer, 2006). Alternatively, another approach is detecting immunogenic peptide sequences using liquid chromatography–mass spectrometry (LC-MS), the so-called MHC-II–associated peptide proteomics (MAPPs) assay (Karle, 2020). In this method, the peptides are naturally processed and presented on MHC-II molecules by in vitro-generated APCs (Purcell et al, 2019). This implies that the MAPPs assay can recreate the whole environmental MHC-II cell machinery and could then be a more suitable tool to detect genuine T cell epitopes than in silico methods or binding assays using synthetic peptides (Fig 1). Thanks to the continuous improvement of MS instrumentation and proteomics software packages, MAPPs assay has attracted more interest from pharmaceutical companies (Karle, 2020). In particular, Spindeldreher et al demonstrated the agreement between MAPPs presented peptides and their potential ability to trigger an immune T-cell activation for ixekizumab and secukinumab (Karle et al, 2016; Spindeldreher et al, 2020). Cassotta et al applied MAPPs to identify MS-eluted ligands from natalizumab (NZM) (Cassotta et al, 2019), a humanized monoclonal antibody targeting α4-integrin on specific B cell clones that were isolated from two patients who revealed clinical immunogenicity. This work demonstrated the attractive predictability power of such an assay. In the last years, the Non-Clinical Immunogenicity Risk Assessment working group of the European Immunogenicity Platform (EIP) (Ducret et al, 2022) and ABIRISK (Rup et al, 2015) consortium have sought to harmonize currently accepted in vitro assays by giving standard accepted criteria systematically applied by industry or research groups.

Most of the MAPPs studies made use of NHS-activated beads or CNBr-activated beads (Sekiguchi et al, 2018; Steiner et al, 2020). The performance of streptavidin magnetic beads on MHC-II-isolated peptides has yet to be shown and compared with commonly used NHS-based beads. Moreover, the current literature does not describe the role of cell medium culture on the morphology and recovery of in vitro-generated APCs in the context of antigen presentation assays such as MAPPs.

In view of a potential breakthrough in MHC-II proteomics immunoaffinity purification, we developed a robust method for identifying the T cell epitopes of therapeutic antibodies with a new bead material for the immuno-enrichment of MHC-II-eluted ligands. In the MAPPs protocol for immunoprecipitation (IP) of peptide–MHC-II complexes, we compared streptavidin-coated magnetic beads with magnetic nanoparticle beads coated with a hydrophilic polymer, polyglycidyl methacrylate (FG magnetic beads) (Sekiguchi et al, 2018). Using this new bead material, it was possible to meaningfully shorten the time of the immunoaffinity step, thus increasing the intrinsic low throughput of the assay. We firstly also evaluated the impact of cell medium composition in the immunophenotyping and morphology of dendritic cells (DCs) that act as APCs with potential implications in MAPPs assay. We used infliximab (IFX), an anti-TNF-α chimeric antibody, as a model therapeutic antibody because it showed a moderate frequency of ADA incidence in clinical practice (Melsheimer et al, 2019; Cohen et al, 2020). We then compared the peptides from the antibody and the total number of MHC-II eluted ligands when using each type of bead. Then, we presented MAPPs analysis of two other well-known immunogenic antigens (Birch pollen allergen, Bet v1a, and KLH) used as positive controls on HLA-genotype donors, experimentally proving the impact of a specific HLA-genotype on the presented MHC-II MS-eluted ligands. Finally, we showed a comparative MAPPs study for early sequence assessment of two investigational biologics in the sequence optimization phase, proving that the MAPPs platform could be helpful in the preclinical immunogenicity screening of new biological entities on a routine basis.

# Results

## Comparison of different culture media in monocytes differentiation into iMo-DCs/mMo-DCs

CD14+ monocytes were isolated from PBMCs with purity around 95% (±3.8%), as determined by flow cytometry (data not shown). To optimize cell culture conditions, monocytes were cultured with GM-CSF and IL-4 in two different media for comparison: one serum-free medium (AIM-V) and one of the most widely used culture medium in research (FBS-supplemented RPMI). The culture media impact on dendritic cell differentiation and maturation was evaluated by analysis of cell morphology, cell viability/recovery, and expression level of surface markers. On day 0, monocytes were rounded in shape, but they slowly changed during differentiation into dendritic cells. After 6 d, the cells showed cytoplasmic protrusions, which were even more pronounced after maturation on day 7, in parallel with the capacity to form cell clusters. Cells cultured in AIM-V appeared strongly adherent and elongated in shape, unlike DCs differentiated in RPMI, which were more rounded and less adherent (Fig 2A). The observed difference in morphology and stickiness shown by the two media might have negatively affected cell recovery and viability in AIM-V. In fact, mature monocyte-derived DC (mMo-DC) viability, evaluated by excluding DAPI+ dead cells, was quite different between cells cultured in the two media: viability was 71% (+6.1%) for cells cultured in AIM-V and 91% (+4.4%) with FBS-supplemented RPMI (Fig 2B). The difference for the cell viability among the two tested media was found to be statistically significant (two tailed $P$-value = 0.000002, $\alpha$ = 0.0001). The mMo-DC recovery, calculated based on the number of initially seeded monocytes, was higher for cells cultured in FBS-supplemented RPMI (29% + 14) than in AIM-V (10% + 3.6) (Fig 2C). For mMo-DC recovery, it was also found to have a statistically significant difference between the two tested media (two tailed $P$-value = 0.00006, $\alpha$ = 0.0001). The low recovery of AIM-V mMo-DCs can hugely affect the number of peptides isolated from dendritic cell membrane detected by LC-MS/MS analysis.

Flow cytometry analysis of cell surface markers showed that in vitro-generated Mo-DCs assumed the typical DC phenotype (Fig 2D). Differentiation of monocytes toward an immature DC phenotype was observed for both tested media by flow cytometry evaluation: monocytes down-regulated the expression of CD14, and Mo-DCs expressed several DC markers involved in the formation of immunological synapse between DCs and (naïve) T cells. These included the costimulatory proteins CD80 and CD86 and the antigen-presenting molecule MHC-II (HLA-DR). Mature Mo-DCs also expressed the DC activation marker, CD83, which was up-regulated accordingly. The main difference observed between the two DC culture media was the expression of CD1a: DCs cultured in AIM-V presented lower levels of CD1a than DCs cultured in FBS-supplemented RPMI.

Moreover, cells cultured in RPMI presented a significantly higher expression of CD80 when compared with AIM-V DCs (Fig 2D). Considering all the obtained data and results, FBS-supplemented RPMI was selected as the most suitable medium for the MAPPs assay to maximize the number of isolated peptides.

## Technical validation of the MAPPs experimental workflow

A technical evaluation of MAPPs experiments was performed to assess the reproducibility of the workflow over time. Preparations from three different healthy donors (three technical replicates each) were incubated with IFX to perform an intradonor, intraday variability. Initially, the samples were independently processed, including cell lysis, peptide extraction, and peptide separation. Each sample was then analyzed using LC-MS/MS.

Results with IFX for the three tested donors (BC01, BC02, and BC03) were plotted as a heatmap for data visualization and comparison (Fig 3A). All three donors showed a characteristic pattern of clusters, some of which covered the hypervariable complementarity-determining (CDR)2 region of the heavy chain (donor BC01) and light chain (donor BC02). Donor BC01 also showed a cluster in the C-terminus of the light chain (L167-184 consensus sequence) and an N-terminus peptide (L6-20, for BC02), whereas BC03 showed a single peptide in the heavy chain CH1 domain. The patterns were very much conserved within each donor, highlighting the robustness of the MAPPs analytical process (Fig 3A, Table S1). Moreover, peptide identification overlap was examined among the three technical replicates of each donor. The number of common peptide sequences across replicates was calculated, resulting in a high degree of reproducibility in MHC-II ligands identification from 1.0–1.5 × $10^6$ mMo-DCs (Fig 3B). The reproducibility of all IFX peptide abundance values was then assessed by the median percentage CV value (% CV, Fig S1). The median CV% of peptide abundances value across replicates was under 25% (median CV% = 22.42) by normalizing on sample total peptide amount. The retention time of all identified common peptides in the three technical replicates was also identical (Figs S2, S3, and S4).

## MHC-II-eluted ligand recovery comparison from two bead types

The difference between the two types of beads (streptavidin versus NHS-activated beads) was evaluated regarding their performance in capturing and isolating MHC-II-eluted ligands.

To compare the two types of beads, experimental evaluations were conducted using DC samples from three different donors for each experimental condition (Fig 4). Statistical analyses were performed to assess any significant difference in the recovery performance between the two types of beads. In particular, the $\log_2$ fold change was calculated as the ratio in abundance or recovery rates of MHC-II ligands between the two bead types. The statistical significance, usually denoted by the $P$-value, reflected the confidence level of the difference observed among the two tested conditions (Fig 4C–E). Points falling above the upper threshold (e.g., $\log_2$ fold change > 1.0 and $-\log_{10}(P$-value) > 1.3) represented significantly up-regulated ligands with higher recovery rates for one bead type compared with the other and vice versa. From a comparison on three tested donors (one out of three showed presented IFX peptides), the recovery of the two beads in terms of total peptide abundance was similar (Fig 4A). A comparable number of over-expressed and down-expressed peptides was found in the three tested donors (Fig 4C–E). On the other hand, a higher number of IFX peptides in the heavy chain (HC) CDR2 region was found in streptavidin magnetic beads-processed samples (four versus two

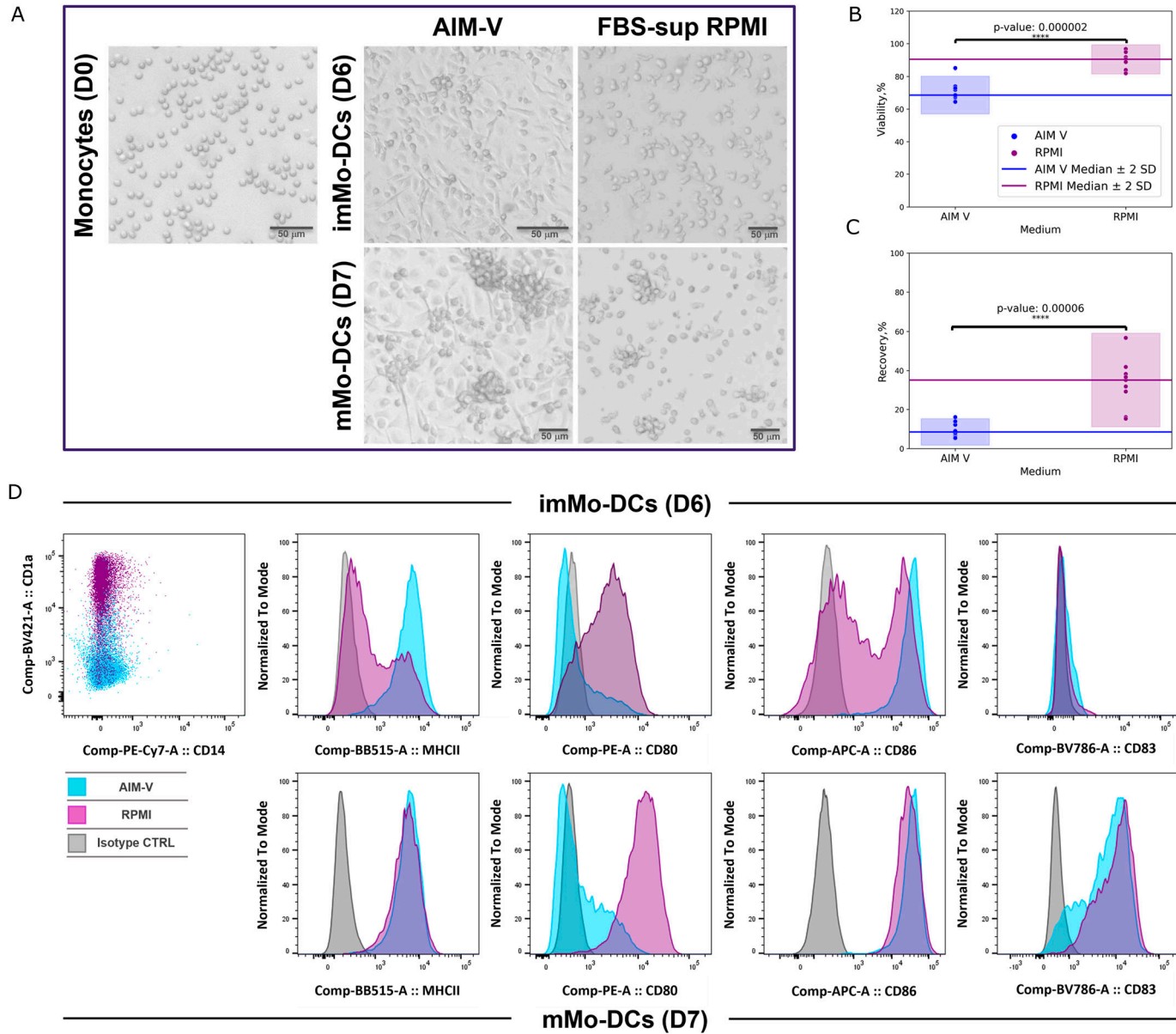

**Figure 2. Comparison of different culture media in monocytes differentiation into iMo-DCs/mMo-DCs.**
**(A)** Representative images of monocytes, immature dendritic cells (imMo-DCs), and mature dendritic cells (mMo-DCs) at day 0 (D0), 6 (D6), and 7 (D7) of culture. Monocytes were cultured with GM-CSF and IL-4 in two different media: serum free media-AIM-V and with FBS-supplemented RPMI. Images were obtained by EVOS M5000 microscope. Magnification: 10×. **(B, C)** Cell viability and recovery in Mo-DC generated with serum-free AIM-V and FBS-supplemented RPMI media. DC viability and recovery were analyzed by NucleoCounter NC200 automated cell counter. **(C)** Cell viability (B) was evaluated by exclusion of DAPI+ dead cells and cell recovery (C) was calculated as a ratio between obtained DCs and seeded monocytes (AIM-V n = 9; RPMI n = 9). The statistical difference among the two media was evaluated using *t* test (****, *P* < 0.0001). **(D)** Immunophenotyping of monocytes, imMo-DCs, and mMo-DCs. Cells were labeled on days 6 (imMo-DCs) and 7 (mMo-DCs) with antibodies specific for the respective markers and analyzed by flow cytometry. Marker expression of DCs cultured with AIM-V and FBS-supplemented RPMI are shown in light blue and violet, respectively. Isotype control labeling is shown in grey. The histograms showed the results from one representative experiment.

peptides in the CDR2 HC cluster and one peptide in the Fc region H403-419 that is missing in the NHS-processed sample, Fig 4B). Moreover, for the common peptides extracted from both bead types, peptide abundance intensities were slightly higher for DCs processed with streptavidin magnetic beads (Fig S5). These results firstly demonstrated the concrete applicability of streptavidin magnetic beads in MAPPs assays.

**MAPPs results on HLA-genotyped donors**

Because the specific HLA-genotype significantly impacts the MHC-II peptidome, we selected five HLA-genotyped donors to be tested in a MAPPs study. MAPPs assay was then performed with IFX biosimilar at 50 μg/ml (Fig 5A). Among the tested donors, four out of five showed FR3 HC peptides (consensus sequence H72-94), and two out

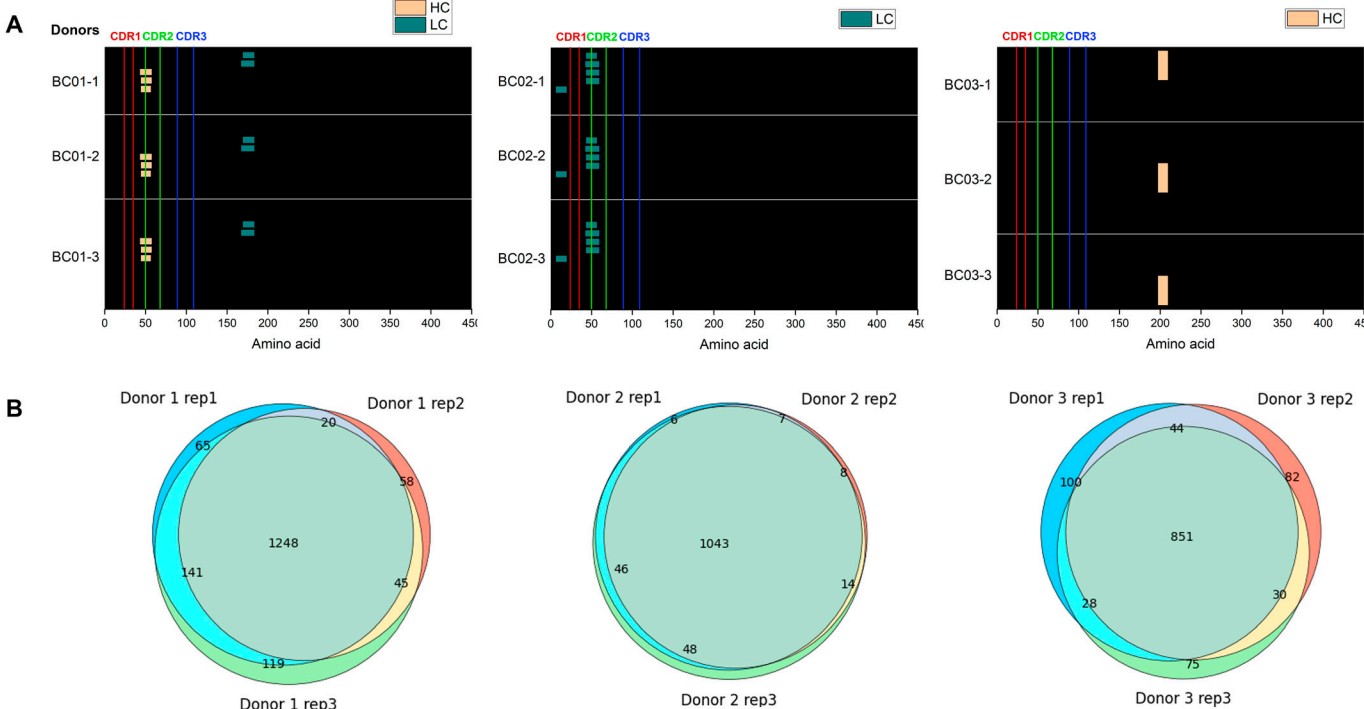

**Figure 3. Technical reproducibility evaluation study.**
Three donors (BC01, BC02, and BC03) were tested in triplicate (independent samples) to assess assay intra-day/intra-donor reproducibility, infliximab (IFX) concentration 10 µg/ml. **(A)** IFX heavy chain peptides (light orange); IFX light chain peptides (dark green). Complementarity-determining regions are shown in different colors according to the legend in the picture. **(B)** Venn plots for reproducibility assessment of technical replicates for peptide sequences. More details are in SI (Figs S2, S3, and S4).

of five the HC CDR3 (H92-106). The donor BC10 gave two peptides on the light chain (LC), L31-45 and L187-202. Finally, the donor BC09 did not provide any peptide. As expected, the genotype differences were reflected in the MS-eluted ligandome. From Fig 5D, BC07 and BC08 donors, sharing allele HLA-DRB1*01:01:01, had a more similar MS-eluted ligand pattern. This result was also confirmed for BC10 and BC11 donors because they shared the allele HLA-DRB1*04:01:01.

In parallel to IFX testing, two other molecules were tested as positive controls, that were KLH and Bet v1a proteins, at 50 µg/ml. This ensured that the entire procedure worked correctly for each donor, from DC maturation to LC-MS analysis. KLH results are reported in Fig 5C. The high antigenicity of the KLH was confirmed by a vast number of peptides and clusters in both isoforms spanning the entire sequence, from N- to C-terminus. Bet v1a heatmap is shown in Fig 5B. Bet v1a antigenicity is less prominent than that of KLH, and it is expected to have a prevalence of around 8–16% positive allergic-type response in the European population (Biedermann et al, 2019). In BC09, BC10, and BC11 donors, a consensus peptide P19-37 was found, spanning the α1 and α2 helices (for BC10, we identified a cluster of three peptides). In the BC09 donor, P2-38 was found in the βI strand and in the α1 and α2 helices (a cluster composed of two peptides), and in BC10 donor P63-77 in the βIV strand. In the BC09 donor, a cluster (P134-160) was found in the C-terminus α3 helix composed of four peptides. Two donors out of five (BC07 and BC08) did not show any presented peptide, both sharing one out of two DRB1 alleles (DRB1*01:01:01).

The binding of a peptide to an MHC-II molecule is primarily driven by a core of nine amino acids, but the location of the 9-mer core within a peptide is not known a priori. Consequently, MHC-II binding data are by nature unaligned concerning the binding core. To identify potential specific binding motifs, the Gibbs clustering algorithm was used to group the input peptide data into a number of clusters based on the optimal local sequence alignment in terms of Kullback–Leiber distance (KLD) (Andreatta et al, 2013). The KLD is an estimation of the information gain of an observed amino acid in a dataset compared with its background distribution (namely its frequency in random protein sequences). This means that the optimal clustering solution is represented by the cluster that has the highest KLD compared with the others. The results obtained on our MS-eluted ligands dataset confirmed the well-known preference of HLA-DR alleles for hydrophobic amino acids in position P1, P4, P9 of the binding core for BC07/BC08 donors sharing DRB1*01:01:01 allele (Fig 6). They also showed a preference for non-bulky amino acids like glycine or alanine in position P6. Furthermore, by looking at identified cluster solutions for BC10-11 sharing DRB1*04:01:01, the distinctive feature was the acidic anchor (aspartate and glutamate) in position P4. Considering experimental epitope length distribution, MHC-II molecules usually accommodate peptides of 13–25 residues in length. Our data seemed aligned with what is known from the literature (Fig 7). Fig 7A reported the overall MHC-II ligandome distribution of all tested molecules on all different donors. The MHC-II epitope length followed a normal distribution, where 15mers and 16mers peptides took over the other peptide lengths.

These findings were confirmed when investigating the individual tested proteins (Fig 7B–D). From violin plots Kernel density distribution, the three investigated proteins had different peptide lengths

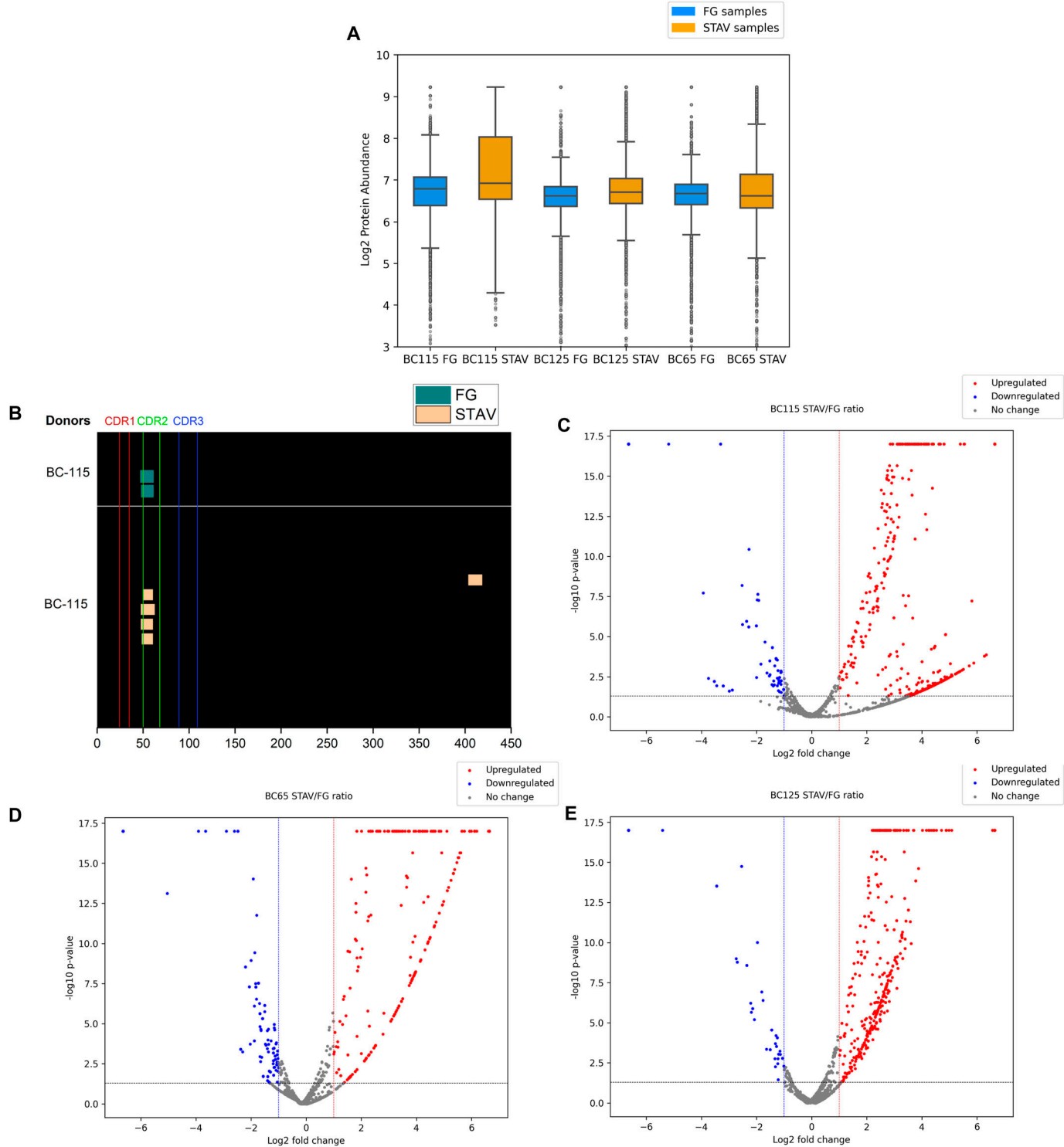

**Figure 4. Evaluation of experimental MHC-II eluted ligands recovery from two types of beads.**
**(A)** Box plot of $\log_2$ summed peptide abundances (named as protein abundance in the plot) comparison between FG-NHS (blue) and streptavidin beads (orange), on three different donors. **(B)** MHC-II-associated peptide proteomics heatmap showing the MHC-II eluted ligands from infliximab (IFX)-treated DCs. Only BC115 (responding to HC-IFX) peptides were shown, instead BC65 and BC125 did not show any IFX peptides. **(C, D, E)** Volcano plots of MHC-II ligand ratio abundance (streptavidin to FG-NHS ratio) for BC115, 65, and 125, respectively. Threshold: $\log_2$ ratio = ±1.0, $-\log_{10}$ P value = 1.3. For details about peptide quantification, please see Fig S7.

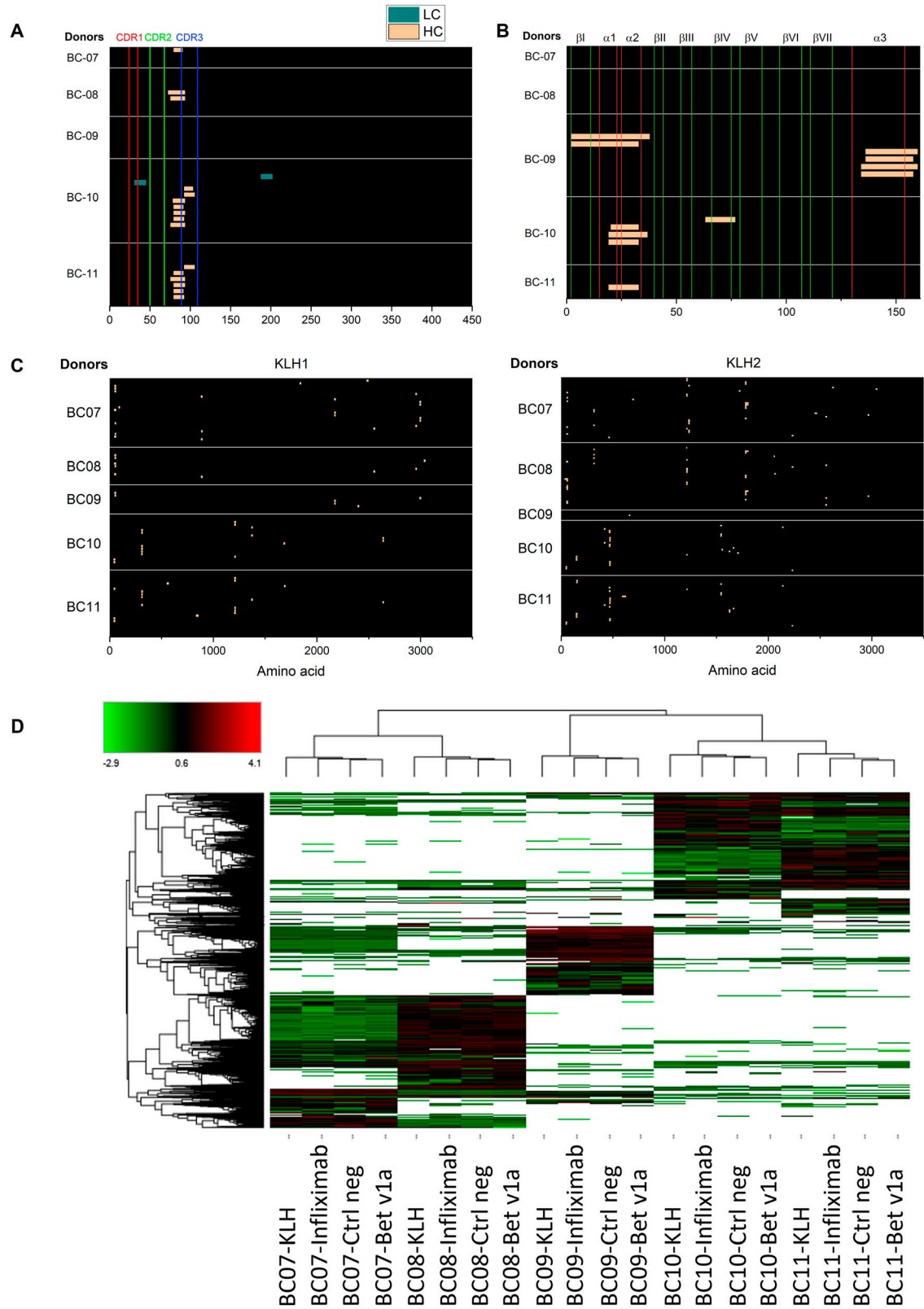

**Figure 5. HLA genotype impact on the pattern of MHC-II-Associated Peptide Proteomics (MAPPs) eluted peptides.**
MAPPs MHC-II peptides of five different HLA-genotyped donors. **(A)** MAPPs MHC-II peptides of infliximab HC and LC. **(B, C)** MAPPs MHC-II peptides of Bet v1a and KLH, respectively. **(D)** Donor clustering based on MHC-II ligandome (red: increased peptide expression; green: decreased peptide expression; black: same peptide expression;

(where wider sections of plots represented a higher probability of a peptide taking a given length, the thinner sections corresponded to a lower probability). On the one hand, for IFX and Bet v1a, a higher percentage frequency for 15mers was obtained (with a median of 16.7% and 50.0%, respectively). On the other side, KLH showed the highest percentage frequency distribution length for 16mers (with a median of 31.0%), followed by 15mers (median of 20.7%).

### MAPPs assay for the risk assessment of drug-induced immunogenicity

The following example elucidated the use of a MAPPs study to collect information that may be used for a preclinical risk sequence assessment of drug-induced immunogenicity. The main goal was to evaluate the immunogenicity risk of two investigational bispecific antibodies to select a lead candidate. These two antibodies are composed of two different HC (herein named HC1 and HC2) heterodimers and a single common LC linked to HC1. HC2 is engineered to have a pair of three CDRs. Moreover, the variable region of the two antibodies contained different sequences in the framework regions (FRs, in particular, differed from some point mutations). However, the two bispecifics shared some common mutations in the CH3 domain of the antibody fragment crystallizable (Fc) region to enable a better yield of the engineered construct. We were also interested in understanding whether the modifications in the HC CH3 domain, compared with a standard trastuzumab IgG1 antibody Fc sequence (ADA rate 4.4% from a recent phase 3 study [Rugo et al, 2021]), may result in the presentation of MHC-II peptides. DC preparations from 10 donors were processed through the MAPPs workflow, and the generated data are shown in Figs 8 and S6. Briefly, the two investigated bispecifics presented the same clusters in the variable regions (e.g., FR3-CDR3, CDR1, and CDR2 clusters for HC2) that were overall found in 4 out of 10 donors. Interestingly, an additional cluster was observed in 70% of tested donors in the Fc domain of the heterodimeric HC1 compared with the homodimeric parental IgG1 structure because of the introduction of non-germline amino acid mutations required for the bispecific production technology. However, this cluster was not considered potentially immunogenic because other companies reported the same bifunctional antibody featuring the same Fc domain construct, eliciting ADAs in only a few patients in clinical trials (Hidalgo et al, 2018). In summary, we assumed that the two bispecific candidates did not present any major flags concerning sequence-driven immunogenicity risk based on MAPPs profiling performed using a pan-HLA-DRB antibody. Likewise, as discussed elsewhere (Di Ianni et al, 2023), MHC-II-presented peptides may also be tolerated in vivo, and further cell-based assays are required to determine potential issues regarding immunogenicity risk assessment (Di Ianni et al, 2023). Nevertheless, MAPPs data, in combination with T cell activation assays and in silico prediction tools, may help to select the most appropriate clinical candidate and/or to optimize moderate/high immunogenic molecules before clinical phases.

## Discussion

Health authorities recommend assessing nonclinical immunogenicity risk with in vitro assays (U.S. Department of Health and Human Services Food and Drug Administration, 2014). MAPPs assay is one of the key in vitro assays in preclinical development used to investigate the antigen processing and presentation of biotherapeutics to the immune system. MAPPs is not considered a high-throughput assay. Therefore, balancing the number of preclinical candidates and the associated workload is crucial. The primary strategy to assess immunogenicity using MAPPs assay should involve a first step where the candidate molecules are screened on a set of HLA-genotyped PBMCs representing a good approximation of the general population (Ducret et al, 2022).

We examined literature data about IFX and Bet v1a to gain some insights into the assay's reliability. Our findings confirmed that the assay can recapitulate known T cell epitopes. Among those, IFX peptides L6-20 and L46-60 generated the highest T-cell response (Hamze et al, 2017). Similarly, the immunodominant epitope Bet v1a P142-156 (resulted in the highest percentage of positive response among 57 tested allergic donors) was also identified in our MAPPs assay (Jahn-Schmid et al, 2005).

This article represented the first report on implementing streptavidin magnetic beads in the MAPPs workflow compared with the standard NHS-activated beads. Our findings suggested that the streptavidin magnetic beads robustly identified the MHC-II eluted ligands. A slightly higher number (four versus two peptides in HC CDR2 and one further peptide in the Fc region that is missing in FG-NHS beads) of IFX peptides was recovered in streptavidin magnetic beads processed samples (Fig 4B). Moreover, when the same IFX peptides were found in samples processed by both bead types, higher intensity peptides were found in extracted ion chromatograms (Fig S5) of streptavidin-processed samples. The total number of MHC-II-eluted ligands was also compared in our beads comparison study. Our results showed that an approximately equal number of over-expressed or down-expressed MHC-II ligands were found from the peptide abundances ratio (streptavidin to FG-NHS beads ratio). Nevertheless, streptavidin magnetic beads have the advantage of a less demanding handling complexity because of a shorter incubation time and a reduced number of steps to be applied in the coating procedure (e.g., 20 h of quenching step is necessary for FG-NHS beads, as opposed to streptavidin magnetic beads that do not require any additional quenching step). In addition, the plausible cross-linking of lysine residues by the N-Hydroxy-succinimide group in the variable region of the HLA-DR antibody used as a coating reagent might hamper the recovery of peptides compared with streptavidin magnetic beads. Another advantage is represented by the high hydrolytic liability of NHS beads, thus needing to perform all bead preparation and immunopurification at 4°C.

In addition, two different cell media were evaluated, and RPMI outperformed AIM-V regarding cell viability (91% versus 71%) and

---

white: missing peptide, Euclidian distance was used for hierarchical clustering). Negative control (Ctrl neg) samples (untreated cells from the same donor) were also included in the MAPPs study.

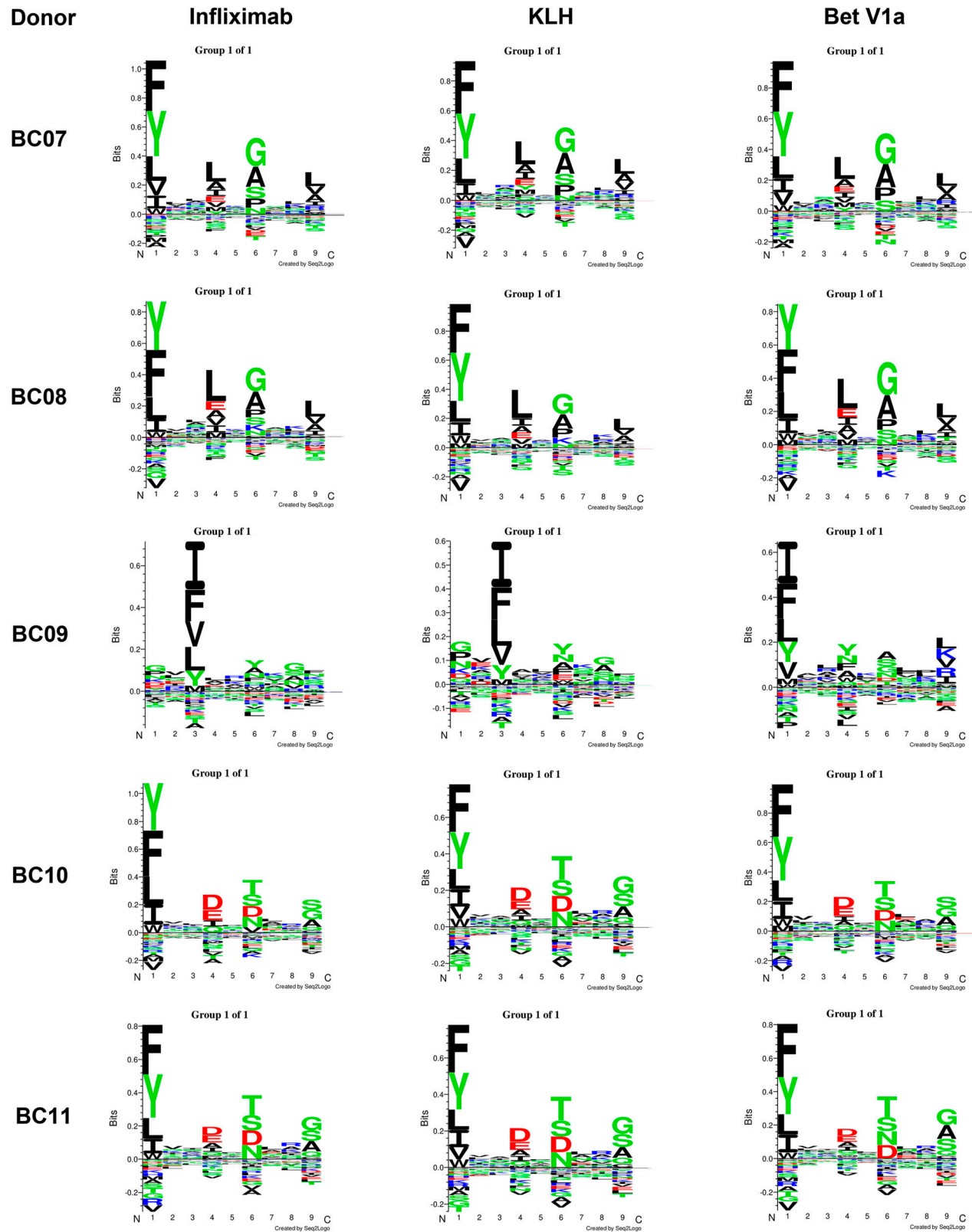

**Figure 6. Gibbs sampling to identify specific binding motifs from MHC-II-Associated Peptide Proteomics peptide dataset.**
Gibbs clustering of MHC-II-eluted ligands of five tested HLA-genotyped donors to identify potential distinctive binding motifs for identified MHC-II-associated peptide proteomics peptides. Information about the HLA genotype of each donor is reported in Table S1.

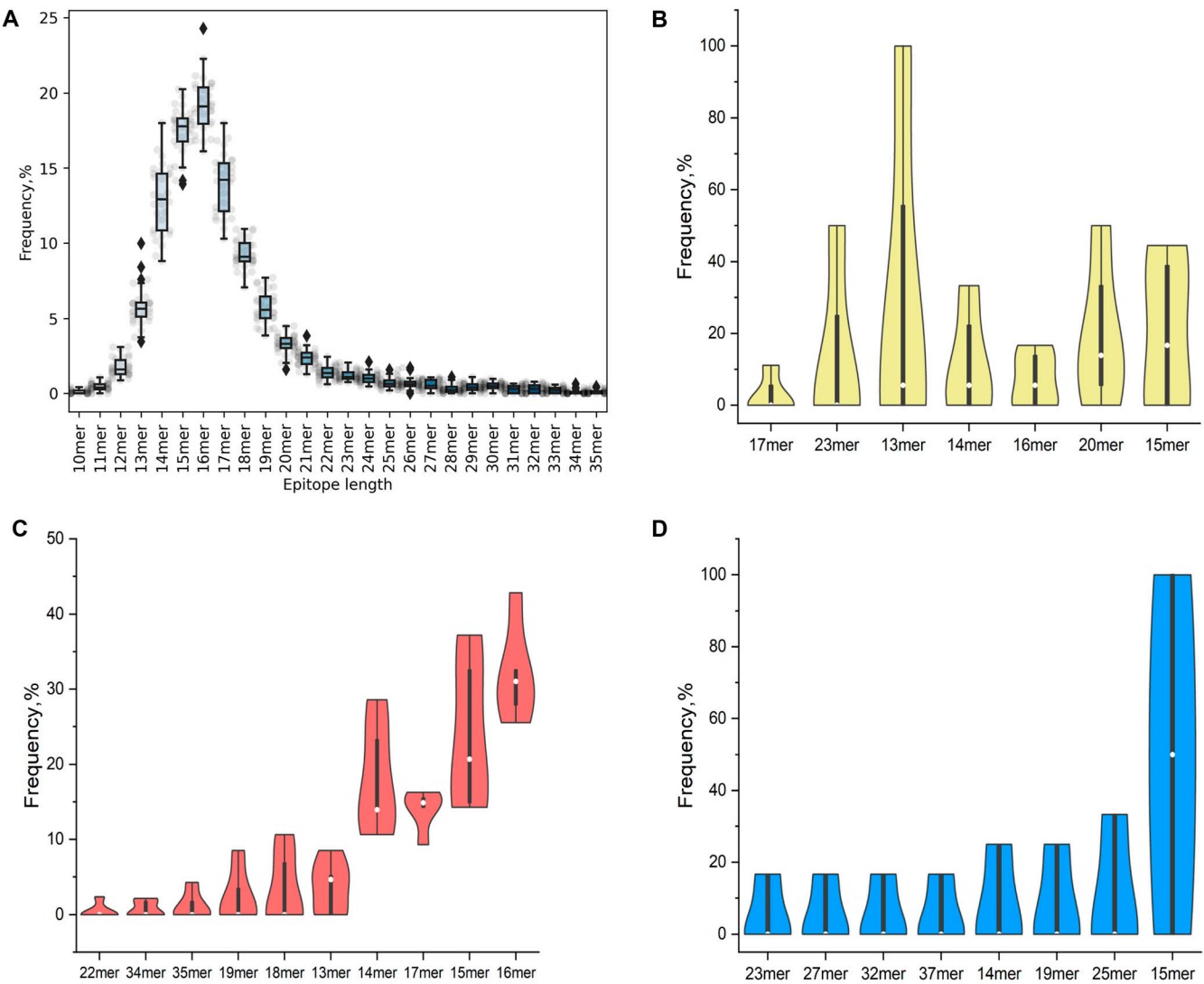

**Figure 7. Results summary about MHC-II–associated peptide proteomics peptide lengths distribution.**
**(A)** Box plot showing a normal distribution for the whole MHC-II–associated peptide proteomics MHC-II–eluted ligands. **(B, C, D)** Violin plots showing the distribution of the data for the different identified peptide lengths in the same five HLA-genotyped donors for (B) infliximab (yellow), (C) KLH (red), and (D) Bet v1a (blue). The median is represented by a white dot in the violin plot. The interquartile range (IQR) is the black bar in the center of violin. The lower/upper adjacent values (the black lines stretched from the bar that overlap with the two ends of Kernel density distribution) are defined as first quartile- 1.5 IQR and third quartile + 1.5 IQR respectively. Peptide lengths were arranged to an ascending order of median percentage frequency for each individual tested protein.

recovery (29% versus 10%). Moreover, the choice of medium greatly impacted the cell surface markers' expression of dendritic cells. In particular, a higher expression of CD1a and CD80 was found in RPMI.

Analysis of antibodies from clinical studies suggests that serious side effects are mainly driven by high levels of IgG antibodies, suggesting a T cell-dependent pathway. As a result, a clinical MAPPs approach would enable a precise patient characterization of presented MHC-II peptides and identify potential T cell epitopes. This would imply the collection of DCs from patients treated with the investigated compounds and analyze the presented MHC-II peptidome via MAPPs assay.

In the future, a clinical MAPPs pipeline may become more established, which will undoubtedly contribute to understanding the preclinical predictive power of this assay. A positive scenario

will help to get insights about undefined immunogenicity mechanisms and develop safer molecules for patients.

# Materials and Methods

### Ethics statement

Buffy coats from healthy human volunteers were supplied by the Azienda Ospedaliero—Universitaria Città della Salute e della Scienza di Torino, Centro di Produzione e Validazione Emocomponenti (Turin, Italy) according to current ethical practices and the samples were anonymized.

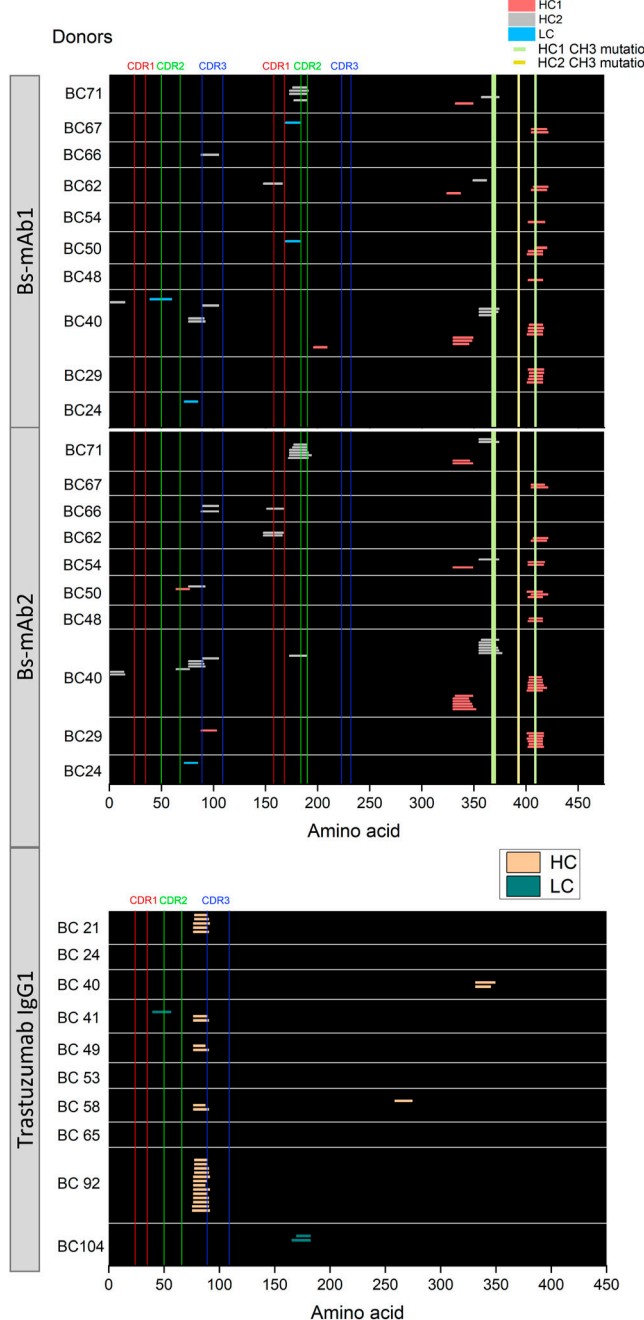

**Figure 8. MHC-II-associated peptide proteomics assay for the risk assessment of drug-induced immunogenicity.**
The two bispecific mAbs (here named Bs-mAb1 and Bs-mAb2) were screened on 10 different donor DC preparations and compared with trastuzumab IgG1 control. KLH was used as positive control (see results in SI, Fig S6). All the molecules were tested at a final concentration of 50 μg/ml.

### PBMCs genotyping

Extracted PBMCs from buffy coats were HLA-genotyped by IMGM Laboratories GmbH by isolating DNA from PBMC samples and subsequently performing high-resolution typing of HLA-DRB1, HLA-DRB3/4/5, HLA-DQA1, HLA-DQB1, HLA-DPA1, and

HLA-DPB1 by long-read next-generation sequencing (LR-NGS, see Table S1).

### Antibodies and proteins evaluated

Model therapeutic antibodies were IFX biosimilar (SIM0006; BioXcell), Bs-mAb1 and Bs-mAb2 were internally produced at Merck KGaA, trastuzumab IgG1 was made at Merck KGaA. Control-positive proteins were KLH (H7017; Sigma-Aldrich) and Bet v1a (BET_1_1A; Biomay).

### In silico analysis of T cell epitopes

In silico analysis of the light chain (LC) and heavy chain (HC) of IFX was carried out on a public database, Immune Epitope Database (IEDB) (Dhanda et al, 2019), using TepiTool (Paul et al, 2016) (http://tools.iedb.org/tepitool) interface for T cell epitope predictions. This is designed as a step-by-step wizard combining MHC class I and II prediction methods. The tool provides recommended default values at each step for predicting and selecting an optimal set of peptides for a given application (26 most frequent panel of alleles, IEDB recommended method). The median of the percentile ranks of the three methods involved was used as consensus percentile ranking (cut-off guidelines: Percentile rank ≤ 10.0). For assessing the most promiscuous peptides, the cut-off was set at 20% as the threshold percentile rank, as suggested by software developers. A summary of the results for IFX and positive controls is reported in the Supporting Information (SI) section (Tables S1, S2, and S3).

### Generation and maturation of monocyte-derived DCs

Human monocyte-derived DCs (mo-DCs) were generated from monocytes obtained from PBMCs of healthy volunteers. PBMCs were purified from buffy coats by density gradient centrifugation on Histopaque (Sigma-Aldrich), and buffy coats were anonymously provided by the "Blood Components Production and Validation Center" (Azienda Ospedaliera Universitaria, Città della Salute e della Scienza, Turin, Italy) for research use only. The CD14+ monocytes were positively selected from PBMCs using anti-CD14-conjugated MicroBeads (Miltenyi Biotec) according to the manufacturer's instructions. To prepare immature Mo-DCs (imMo-DC), the enriched monocyte fraction was incubated in six-well culture plates ($6 \times 10^6$ cells/5 ml/well) with two different culture media supplemented with 200 ng/ml of recombinant human GM-CSF and 50 ng/ml of recombinant human IL-4 (both Miltenyi Biotec) for 6 d at 37°C in 5% $CO_2$. The two culture media tested in this study were: serum-free AIM-V medium (Gibco) and Roswell Park Memorial Institute Medium (RPMI) 1,640 (Gibco) supplemented with 10% heat-inactivated FBS, 100 U/ml penicillin, 100 μg/ml streptomycin, 2 mM glutamax, 1 mM sodium pyruvate, and 1× MEM nonessential amino acids (all from Gibco). On day 2, the half culture medium was replaced by fresh medium containing twice the concentration of both GM-CSF and IL-4 cytokines compared with the initial concentration. On day 6, imMo-DCs were resuspended in fresh FBS-containing medium supplemented or not with different concentrations of IFX, KLH, and Bet v1a and incubated for 3 h at 37°C in 5% $CO_2$. 100 ng/ml of LPS from *Escherichia coli* O55:B5 (Sigma-Aldrich, Merck) was added to the cell culture to induce dendritic cell maturation. After 24 h, mature Mo-DCs (mMo-DCs) were harvested and their immunophenotype

analyzed by flow cytometry. Cell pellets were frozen at −80°C for at least 16 h before proceeding with the cell lysis step.

## Cell morphology, viability, and immunophenotypic analysis

Images of monocytes, imMo-DCs, and mMo-DCs were captured by EVOS M5000 light microscope with phase contrast (Thermo Fisher Scientific). Cell recovery and viability were analyzed by Nucleo-Counter NC200 automated cell counter (ChemoMetec). ImMo-DCs and mMo-DCs were labeled with fluorochrome-conjugated monoclonal antibodies for 30 min at 4°C after blocking nonspecific sites with Human BD Fc Block (BD Biosciences) to detect expression of maturation markers and co-stimulatory molecules. The following mAbs were used: anti-CD14-PECy7, anti-CD80-PE, anti-CD83-BV786, anti-CD86-APC, anti-MHC class II antigen-BB515 (BD Biosciences), anti-CD1a-eFluor450 (Thermo Fisher Scientific). Nonspecific antibody binding was assessed using appropriate isotype controls (mouse IgG2a PE-Cy7, mouse IgG2a PE, mouse IgG1 BV786, mouse IgG1 APC, mouse IgG2a BB515, mouse IgG1 eFluor450). Cell debris and dead cells were excluded from the analysis based on light-scatter properties and 7-AAD fluorescence signal (BD Biosciences). Immunophenotype (IPT) was assessed by BD FACSCelesta SORP (BD Biosciences), and data analysis was carried out by FlowJo software (BD Biosciences).

## Cell pellet extraction procedure

Frozen pellets were thawed, and mature DCs were lysed in ice-cold lysis buffer (20 mM Tris buffer pH 7.8) containing 5 mM $MgCl_2$, 1% Triton X-100 (11332481001; Roche Diagnostics), and one tablet of cOmplete Mini protease inhibitors (11836153001; Roche Diagnostics) for 1 h at 4°C. The volume was scaled accordingly to the cell pellet number. After centrifugation, the lysate was incubated with two different immunoenrichment reagents: (i) biotinylated anti-HLA-DR antibody (clone G46-6, 307614; BioLegend) coupled to streptavidin sepharose magnetic beads (28985738; Cytiva, GE Healthcare), (ii) anti-HLA-DR antibody (clone G46-6, Purified NA/LE Mouse Anti-Human HLA-DR, 555809; New England BioLabs GmbH) coupled to polyglycidyl methacrylate FG-NHS magnetic beads (TAS8848N1141; Tamagawa Seiki Co, Ltd.). FG beads were prepared accordingly to vendor protocol E105. The comparability between the two bead preparations was ensured by the equimolar quantity of the capturing reagent present in the two preparations for the same amount of beads. After overnight incubation at 4°C, beads were washed several times with PBS and PBS containing 0.1% Zwittergent 3–12 (693015; Merck KGaA). After washing, peptides were eluted from the beads by adding 25 μl of an aqueous solution containing 2% vol/vol ACN 0.05% vol/vol trifluoroacetic acid two times at 37°C on a thermomixer for 30 min. Eluates were pooled and injected into the LC-MS instrument.

## LC-MS method

MHC-II peptide preparations obtained from matured monocyte-derived DCs (Mo-DCs) samples were separated on a nanocapillary liquid chromatography system (UltiMate 3000 RSLC; Thermo Fisher Scientific) using a C18 reversed-phase column (75 μm i.d. × 150 mm, Pep Map RSLC C18, 2 μm, 100 Å, set at 35°C, ES904; Thermo Fisher Scientific) connected to a Q-Exactive Plus Orbitrap mass

spectrometer (Thermo Fisher Scientific) via nanoelectrospray ionization (EasySpray source; Thermo Fisher Scientific). Samples (20 μl volume dissolved in 0.05% (vol/vol) trifluoroacetic acid in 2% (vol/vol) acetonitrile/water were loaded for 4 min at 10 μl/min onto an Acclaim PepMap C18 trap column (300 μm i.d. × 5 mm, 160454; Thermo Fisher Scientific). Peptides were then eluted at a flow rate of 400 nl/min using a linear 50 min gradient of 2–40% B, followed by a 10 min column wash at 95% B, and re-equilibration for 19 min (buffer A: 0.1% [vol/vol] formic acid in water; buffer B: 0.1% [vol/vol] formic acid in acetonitrile). MHC-II peptides were analyzed by tandem MS using standard operating parameters. Survey scans (scanning range 266.7–2,000 m/z) were recorded in the Orbitrap mass analyzer at a resolution of 70,000 at 200 m/z, max injection time 100 ms, AGC target $3 \times 10^6$ ions, one microscan, without the lock mass option enabled. Data-dependent MS/MS spectra of the 10 most abundant ions from the survey scan were acquired in the HCD Orbitrap cell at a resolution of 17,500 at 200 m/z, max injection time 100 ms, AGC target $1 \times 10^5$ ions, one microscan, isolation window 2 m/z. Target ions selected for MS/MS were excluded with a dynamic exclusion of 10 s.

## Data analysis

Peptides were identified with the SEQUEST HT search algorithm against the SWISS-PROT human database containing the sequence of the evaluated protein using Proteome Discoverer software ver. 2.5.0.400 (Thermo Fisher Scientific). The search was performed with a mass tolerance of ±10 ppm for precursor ions and ±0.02 D for fragment ions. Met-sulfoxide, Asn/Gln deamidation, and N-terminal pyroglutamylation were considered as variable modifications. Data were searched without enzyme specificity. Peptides with a delta mass of <10 ppm (for precursor ions) to the expected mass, cross-correlation values (Xcorr) > 1.9 for doubly charged ions, >2.3 for triply charged ions, and >2.6 for quadruply charged ions were considered as true hits. Manual inspection of MS fragment spectra was also performed to reinforce the assignment of each peptide.

Label-free quantitation (LFQ) approach was used to statistically test differences among experimental conditions by Proteome Discoverer using *t* test background based hypothesis test (Navarro et al, 2014). Firstly, this was applied to evaluate the intra-donor and intra-assay technical reproducibility. Secondly, it enabled the selection of the optimal experimental settings to ensure that the method reported the maximum number of MHC-II ligands.

The Proteome Discoverer LFQ processing workflow contained additional nodes compared with the basic Search workflow with SEQUEST HT. More details about the different nodes used for LFQ analysis are shown in SI (Fig S7).

Further statistical data were obtained by standard Proteome Discoverer tools (Box-Whiskers plots, volcano plots, and hierarchical clustering). In particular, for hierarchical clustering, Euclidean distance ($d_{AB}$) was used as a metric (Equation (1)):

$$d_{AB} = \sqrt{\sum_{i=1}^{n}(e_{Ai} - e_{Bi})^2} \qquad (1)$$

where $d_{AB}$ is the distance between donors A and B, and the $e_{Ai}$ and $e_{Bi}$ are expression values of the $i^{th}$ peptide for donors A and B.

### MHC-II peptide alignment and clustering using a Gibbs sampling approach

Gibbs clustering method was applied to our unaligned peptides dataset of variable length of HLA-genotyped donors to deconvolute HLA-specific binding motifs using GibbsCluster - 2.0 online web server (GibbsCluster 2.0 - DTU Health Tech - Bioinformatics Tools), using recommended configurations for MHC-II ligands (Andreatta et al, 2013). Each cluster is represented by a position-specific scoring matrix, and the sampling method aims at maximizing the information content (expressed as bits) of individual matrices, whereas minimizing the overlap between distinct clusters. For each cluster size, the solution with the highest KLD score was recorded (Andreatta et al, 2013, 2017).

## Data Availability

The mass spectrometry proteomics data have been deposited to the ProteomeXchange Consortium via the PRIDE partner repository with the dataset identifier PXD037958. The data are also available from the corresponding author upon request.

## Supplementary Information

## Acknowledgements

We would like to thank Valeria Castagna for cellular training and support in the early phase of the project and Vanita Sood and Klaus Urbahns for the article revision. The authors would also like to thank anonymous reviewers for their thorough and insightful comments that helped improve the article. This research was funded by the healthcare business of Merck KGaA, Darmstadt, Germany.

### Author Contributions

A Di Ianni: conceptualization, data curation, software, investigation, methodology, and writing—original draft, review, and editing.
T Fraone: conceptualization, data curation, software, investigation, methodology, and writing—original draft, review, and editing.
P Balestra: conceptualization, supervision, and writing—review and editing.
K Cowan: conceptualization, supervision, funding acquisition, project administration, and writing—review and editing.
F Riccardi Sirtori: conceptualization, supervision, funding acquisition, project administration, and writing—review and editing.
L Barbero: conceptualization, data curation, supervision, methodology, project administration, and writing—original draft, review, and editing.

### Conflict of Interest Statement

The authors declare the following financial interests/personal relationships, which may be considered potential competing interests: T Fraone, P Balestra, K Cowan, F Riccardi Sirtori, and L Barbero are employees of Merck KGaA.

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
