## [Reviewer comments · Life Science Alliance]

Assessing MAPPs assay as a tool to predict the immunogenicity potential of protein therapeutics

Tiziana Fraone, Piercesare Balestra, Kyra Cowan, Federico Riccardi Sirtori, Luca Maria Barbero and Andrea Di Ianni
DOI: <https://doi.org/10.26508/lsa.202302095>

Corresponding author(s): Dr. Andrea Di Ianni (University of Turin) and Luca Maria Barbero (Merck KGaA)

Review Timeline:

Submission Date:	2023-04-17
Editorial Decision:	2023-06-03
Revision Received:	2023-08-02
Editorial Decision:	2023-08-31
Revision Received:	2023-09-22
Editorial Decision:	2023-09-27
Revision Received:	2023-10-02
Accepted:	2023-10-03

Scientific Editor: Novella Guidi

Transaction Report:

June 3, 2023

Re: Life Science Alliance manuscript #LSA-2023-02095-T

Dr. Andrea Di Ianni
University of Turin
NBE-DMPK Innovative BioAnalytics, Merck Serono RBM S.p.A., an affiliate of Merck KGaA, Darmstadt,
Germany, Via Ribes 1, 10010 Colletterto Giacosa (TO), Italy
Via Ribes 1
Colletterto Giacosa, Italy/Turin 10010
Italy

Dear Dr. Di Ianni,

Thank you for submitting your manuscript entitled "Assessing MAPPs assay as a tool to predict immunogenicity potential of protein therapeutics". The manuscript has been evaluated by expert reviewers, whose reports are appended below. Unfortunately, after an assessment of the reviewer feedback, our editorial decision is against publication in Life Science Alliance.

Although your manuscript is intriguing, I feel that the points raised by the reviewers are more substantial than can be addressed in a typical revision period. If you wish to expedite publication of the current data, it may be best to pursue publication at another journal.

Given the interest in the topic, I would be open to re-submission to Life Science Alliance of a significantly revised and extended manuscript that fully addresses the reviewers' concerns and is subject to further peer review. If you would like to resubmit this work to Life Science Alliance, you may submit an appeal directly through our manuscript submission system. Please note that priority and novelty would be reassessed at re-submission.

Regardless of how you choose to proceed, we hope that the comments below will prove constructive as your work progresses.

Thank you for thinking of Life Science Alliance as an appropriate place to publish your work.

Sincerely,

Reviewer #1 (Comments to the Authors (Required)):

In the publication "Assessing Major Histocompatibility Complex-Associated Peptide Proteomics assay as a tool to predict immunogenicity potential of protein therapeutics and antigens" by Di Ianni et al. authors developed a method which helps to identify the T cell epitopes of therapeutic antibodies. Authors utilized monocyte derived DC as model and source of MHC II antigens coming from biologics. Overall work appears to be important in the aspect of growing biologics significance in modern therapies.

Major points:

1. Please provide PRM/MRM/SRM data to confirm identified peptides, peptides quantities vs controls.
2. Please provide the information about the fraction of the peptide of interest (PSM) vs total peptides extracted from MHC II.

3. Please provide information about reproducibility of sequence identification in the different experiments in the table format.
 4. Please generate sequence logo plots for all identified, eluted peptides (MHC specific)
 5. Please indicate clearly number of replicates -technical and biological.
 6. Please provide overlap Venn Diagrams or Correlation plots using peptide/protein intensity etc. of all extracted peptide sequences/protein across different donors and replicates.
- Minor points:
1. Figure numbering in figures part missing.

Reviewer #2 (Comments to the Authors (Required)):

The manuscript entitled "Assessing Major Histocompatibility Complex-Associated Peptide Proteomics assay as a tool to predict immunogenicity potential of protein therapeutics and antigens" reports on recent improvements in methods for isolation of MHC-II peptides to be analyzed in the MHC-II-associated peptide proteomics using mass spectrometry. The MAPPs assay is important to identify peptides that can be used for vaccine development or that should be avoided when developing new biotherapeutics drug candidates.

The manuscript is relatively difficult to read and the use of the English language is sometimes quite uncertain. It lacks scientific rigor and the examples provided do not comply with common guidance with respect to number of donor tested and method of analysis. In addition, the literature citations are quite outdated and recent reviews would provide a helpful framework to guide the authors on the use of the MAPPs assay in the context they present. In particular, the following citations need to be added in the manuscript:

-Assay format diversity in pre-clinical immunogenicity risk assessment: Toward a possible harmonization of antigenicity assays.

Ducret A, Ackaert C, Bessa J, Bunce C, Hickling T, Jawa V, Kroenke MA, Lamberth K, Manin A, Penny HL, Smith N, Terszowski G, Tourdot S, Spindeldreher S. *MABs*. 2022 Jan-Dec;14(1):1993522. doi: 10.1080/19420862.2021.1993522.

Applying MAPPs Assays to Assess Drug Immunogenicity.

Karle AC. *Front Immunol*. 2020 Apr 21;11:698. doi: 10.3389/fimmu.2020.00698. eCollection 2020. PMID: 32373128

Enabling Routine MHC-II-Associated Peptide Proteomics for Risk Assessment of Drug-Induced Immunogenicity. Steiner G, Marban-Doran C, Langer J, Pimenova T, Duran-Pacheco G, Sauter D, Langenkamp A, Solier C, Singer T, Bray-French K, Ducret A. *J Proteome Res*. 2020 Sep 4;19(9):3792-3806. doi: 10.1021/acs.jproteome.0c00309. Epub 2020 Aug 25. PMID: 32786679

Overall, this manuscript presents only two novel aspects.

1) The use of magnetic streptavidin beads to capture biotinylated HLA-DR antibodies - the authors demonstrate some data highlighting the improvement of the methods but only show results for two donors in duplicate, which does not provide a sound statistical basis. In addition, the results would have been more convincing when using the endogenous peptidome results as well.

2) The authors present an intriguing method to calculate an apparent saturation binding experiment. At first look, this looks interesting. However, there are plenty of caveats that most likely would invalidate the calculations. A first argument is that the authors need to demonstrate that the amount of  produced and presented is directly proportional to the amount of the therapeutics they incubate to the DCs. In particular, the internalization of the therapeutics, its degradation in the endosome compartment, loading onto the MHC-II receptor and trimming by cathepsins might not be linear. Second, there is also an equilibrium of the MHC-II peptide receptor complex to dissociate during the time of the sample preparation. This is clearly not accounted for in the calculations. Finally, MS quantification using LFQ is not been robustly used for single peptide in the absence of an internal standard. Hence, the linearity of each peptide's response is not warranted. The authors should at minimum correlate their data with estimated binding constant as for example generated using the package NetMHCPanII.

In my view, the authors show a number of results but they fail to discuss the relevance of them in the context of the analysis they perform. For example, it would have been of genuine interest to investigate whether the original Infliximab molecule and the biosimilar provided identical clusters in this context, and to compare results across the many studies that have looked into infliximab.

I have a number of general remarks, which are overall too numerous to list here but a few:

- lines 34/35: immunogenic biotherapeutics may give rise to ADAs directly from the 1st injection, although this is not the usual rule. However, this should be worded differently. Also, sometimes, the repeated administration of the therapeutics may also induce tolerance.

- lines 44-68: This whole section would benefit of a complete rewriting following the lecture of recent publications
- Line 63: reference 25 is not the only publication on MAPPs.
- Line 174-221: The data analysis section is poorly written and do not allow replication of the analysis. FDR calculation is not compatible with filtering with Xcorr/DCn values. Also, the LFQ node needs better description.
- Line 260-278: there cannot be a relevant reproducibility analysis with 2 donors investigated in duplicate.
- Line 292-314: Please consider here that you evaluate peptide abundance and not protein abundance. Please correct.
- Line 337: the table is Table S1
- Lines 355-367: this is not relevant to the result section.
- Line 470-472: It is quite clear that the authors had already a go to publish their paper?
- Line 531: reference 19 and 22 are identical

Reviewer #1 (Comments to the Authors (Required)):

In the publication "Assessing Major Histocompatibility Complex-Associated Peptide Proteomics assay as a tool to predict immunogenicity potential of protein therapeutics and antigens" by Di Ianni et al. authors developed a method which helps to identify the T cell epitopes of therapeutic antibodies. Authors utilized monocyte derived DC as model and source of MHC II antigens coming from biologics. Overall work appears to be important in the aspect of growing biologics significance in modern therapies.

Major points:

1. Please provide PRM/MRM/SRM data to confirm identified peptides, peptides quantities vs controls. **No PRM/MRM/SRM data was acquired. The method acquired a first survey scan on the precursor ions of intact peptide masses, and then the top10 ions were injected in the High Collisional Dissociation Cell of the Orbitrap Q-Exactive Plus to fragment the parent ions and get the complete sequencing at amino acid level.**
2. Please provide the information about the fraction of the peptide of interest (PSM) vs total peptides extracted from MHC II. **PSM acronym stands for Peptide Spectra Matches, namely the same peptide can have different PSMs according to the different charges that the same peptide can take in mass spec. Could you please spell it out a little bit? Do you want us to explain the fraction of pulsed-molecule presented peptides PSMs vs the total amount of PSMs?**
3. Please provide information about reproducibility of sequence identification in the different experiments in the table format. **Table S1, with peptide sequence in table format for the three different tested donors and Figure 3 for technical reproducibility evaluation.**
4. Please generate sequence logo plots for all identified, eluted peptides (MHC specific). **Done, see figure 6.**
5. Please indicate clearly number of replicates -technical and biological. **Please refer to Figure 3 and paragraph "Technical validation of the MAPPs experimental workflow".**
6. Please provide overlap Venn Diagrams or Correlation plots using peptide/protein intensity etc. of all extracted peptide sequences/protein across different donors and replicates. **Please refer to Figure 3**

Minor points:

1. Figure numbering in figures part missing.

Reviewer #2 (Comments to the Authors (Required)):

The manuscript entitled "Assessing Major Histocompatibility Complex-Associated Peptide Proteomics assay as a tool to predict immunogenicity potential of protein therapeutics and antigens" reports on recent improvements in methods for isolation of MHC-II peptides to be analyzed in the MHC-II-associated peptide proteomics using mass spectrometry. The MAPPs assay is important to identify peptides that can be used for vaccine development or that should be avoided when developing new biotherapeutics drug candidates. The manuscript is relatively difficult to read and the use of the English language is sometimes quite uncertain. It lacks scientific rigor and the examples provided do not comply with common guidance with respect to number of donor tested and method of analysis. In addition, the literature citations are quite outdated and recent reviews would provide a helpful framework to guide the authors on the use of the MAPPs assay in the context they present. In particular, the following citations need to be added in the manuscript:

-Assay format diversity in pre-clinical immunogenicity risk assessment: Toward a possible harmonization of antigenicity assays.

Ducret A, Ackaert C, Bessa J, Bunce C, Hickling T, Jawa V, Kroenke MA, Lamberth K, Manin A, Penny HL, Smith N, Terszowski G, Tourdot S, Spindeldreher S. MAbs. 2022 Jan-Dec;14(1):1993522. doi: 10.1080/19420862.2021.1993522.

Applying MAPPs Assays to Assess Drug Immunogenicity.

Karle AC. Front Immunol. 2020 Apr 21;11:698. doi: 10.3389/fimmu.2020.00698. eCollection 2020. PMID: 32373128

Enabling Routine MHC-II-Associated Peptide Proteomics for Risk Assessment of Drug-Induced

Immunogenicity.

Steiner G, Marban-Doran C, Langer J, Pimenova T, Duran-Pacheco G, Sauter D, Langenkamp A, Solier C, Singer T, Bray-French K, Ducret A.J Proteome Res. 2020 Sep 4;19(9):3792-3806. doi: 10.1021/acs.jproteome.0c00309. Epub 2020 Aug 25. PMID: 32786679

The previous citations were introduced in the new introduction.

Overall, this manuscript presents only two novel aspects.

1) The use of magnetic streptavidin beads to capture biotinylated HLA-DR antibodies - the authors demonstrate some data highlighting the improvement of the methods but only show results for two donors in duplicate, which does not provide a sound statistical basis. In addition, the results would have been more convincing when using the endogenous peptidome results as well.

2) The authors present an intriguing method to calculate an apparent saturation binding experiment. At first look, this looks interesting. However, there are plenty of caveats that most likely would invalidate the calculations. A first argument is that the authors need to demonstrate that the amount of

produced and presented is directly proportional to the amount of the therapeutics they incubate to the DCs. In particular, the internalization of the therapeutics, its degradation in the endosome compartment, loading onto the MHC-II receptor and trimming by cathepsins might not be linear. Second, there is also an equilibrium of the MHC-II peptide receptor complex to dissociate during the time of the sample preparation. This is clearly not accounted for in the calculations. Finally, MS quantification using LFQ is not been robustly used for single peptide in the absence of an internal standard. Hence, the linearity of each peptide's response is not warranted. The authors should at minimum correlate their data with estimated binding constant as for example generated using the package NetMHCpanII. →The quantitation part of different concentration was ruled out for the revised manuscript.

In my view, the authors show a number of results but they fail to discuss the relevance of them in the context of the analysis they perform. For example, it would have been of genuine interest to investigate whether the original Infliximab molecule and the biosimilar provided identical clusters in this context, and to compare results across the many studies that have looked into infliximab.

I have a number of general remarks, which are overall too numerous to list here but a few:

- lines 34/35: immunogenic biotherapeutics may give rise to ADAs directly from the 1st injection, although this is not the usual rule. However, this should be worded differently. Also, sometimes, the repeated administration of the therapeutics may also induce tolerance. Corrected in the intro, rephrased in the new introduction...

- lines 44-68: This whole section would benefit of a complete rewriting following the lecture of recent publications See new introduction

- Line 63: reference 25 is not the only publication on MAPPs. Please see new references for MAPPs in the revised introduction.

- Line 174-221: The data analysis section is poorly written and do not allow replication of the analysis. FDR calculation is not compatible with filtering with Xcorr/DCn values. Also, the LFQ node needs better description. All the different LFQ nodes with our custom settings were explained in details in Supplemental material.

- Line 260-278: there cannot be a relevant reproducibility analysis with 2 donors investigated in duplicate. The technical validation was now assessed on three different donors in triplicate (n=9)

- Line 292-314: Please consider here that you evaluate peptide abundance and not protein abundance. Please correct. Corrected.

- Line 337: the table is Table S1

- Lines 355-367: this is not relevant to the result section. It was eliminated from the Results section

- Line 470-472: It is quite clear that the authors had already a go to publish their paper? This is simply a sentence where the reader can access to our MAPPs dataset on ProteomeXchange PRIDE repository.

- Line 531: reference 19 and 22 are identical. Modified in the rephrased and revised introduction.

August 31, 2023

Re: Life Science Alliance manuscript #LSA-2023-02095-TR-A

Dr. Andrea Di Ianni
University of Turin
NBE-DMPK Innovative BioAnalytics, Merck Serono RBM S.p.A., an affiliate of Merck KGaA, Darmstadt, Germany, Via Ribes 1,
10010 Colletterto Giacosa (TO), Italy
Via Ribes 1
Colletterto Giacosa, Italy/Turin 10010
Italy

Dear Dr. Di Ianni,

Thank you for submitting your revised manuscript entitled "Assessing MAPPs assay as a tool to predict immunogenicity potential of protein therapeutics" to Life Science Alliance. The manuscript has been seen by the original reviewers whose comments are appended below. While the reviewers continue to be overall positive about the work in terms of its suitability for Life Science Alliance, some important issues remain.

Our general policy is that papers are considered through only one revision cycle; however, given that the suggested changes are relatively minor, we are open to one additional short round of revision. Please note that I will expect to make a final decision without additional reviewer input upon re-submission.

Please submit the final revision within one month, along with a letter that includes a point by point response to the remaining reviewer comments.

To upload the revised version of your manuscript, please log in to your account: <https://lsa.msubmit.net/cgi-bin/main.plex>
You will be guided to complete the submission of your revised manuscript and to fill in all necessary information.

B. MANUSCRIPT ORGANIZATION AND FORMATTING:

Sincerely,

Reviewer #2 (Comments to the Authors (Required)):

The revised manuscript entitled "Assessing Major Histocompatibility Complex-Associated Peptide Proteomics assay as a tool to predict immunogenicity potential of protein therapeutics and antigens" reports on recent improvements in methods for isolation of MHC-II peptides to be analyzed in the MHC-II-associated peptide proteomics using mass spectrometry. The MAPPs assay is important to identify peptides that can be used for vaccine development or that should be avoided when developing new biotherapeutics drug candidates.

Despite some improvements on the content and a more logical organization of the manuscript, the manuscript remains relatively difficult to read. The authors have removed from their publications one of the controversial aspects of this work, namely, on their attempt to use single peptide quantification with LFQ. However, there are still some imprecisions regarding results and additional clarification is required especially with regards to the interpretation of some of the results. In addition, except for the requested citations, the literature has not been updated to more recent publications, and this does not reflect positively on the presented work.

Please note that Table S1 was NOT included in the revised submission. Some of the comments I will be making might therefore be due to the lack of this information.

Line 46, reference 13, 14: It would be more proper to quote the original document from the EMA and FDA. Also, please note that the FDA 2014 guideline is now accompanied by a draft guidance from 2022 - please add reference and include modifications as needed.

Line 89 - "Compared to previous published MAPPs, we developed a robust method ...". Steiner et al. (reference 38 in the publication) already showed that MAPPs protocols are robust and reproducible. Please amend the statement.

Line 98: high frequency - please define what is high frequency in this context. Reference 1 is outdated in this respect and there are more recent data to indicate that Infliximab is more on the moderate side regarding ADA incidence.

Line 155: Please indicate how mMoDCs were frozen as this is mentioned in line 171 of the manuscript.

Line 191: The column is not self-packed as it bears a catalog number from ThermoFisher. Please correct.

Line 200: Survey scans m/y 266.7 to 4000 m/z: are you certain of the upper range? This would make the instrument very, very slow. Please check.

Line 217 to 225 (and SI): please do not describe in detail the LFQ data package. It is more appropriate to put in the SI the graphical representation of the nodes as provided by Proteome Discoverer. If desired, you can add screenshots of each of the node.

Line 239: reference 42 is outdated, please add the 2017 reference mentioned on the website.

Lines 257-259 / Lines 259-261: These two sentences appear to be redundant?

Lines 276 and following: please number your donors in a consistent manner. At present, there are three different numerations, and this is confusing to the reader.

Line 291 - $1.0-1.5 \times 10^6$ DCs: in which stage were these DCs counted? iMo-DCs, mMo-DCs, monocyte? Please precise.

Line 295: Figure 3C is redundant to Figure S2, S3 and S4. In my opinion, Figure 3C is not absolutely required in the main body of the text.

Line 300: three different donors were tested, were these donors tested in several technical replicates? There can be no statistics out of $n=1$ sample. Please precise.

Line 308: Figure 4A would benefit from a \log_2 representation on the y axis rather than a \log_{10} .

Lines 308-314: There is an apparent contradiction here. In Fig. 4a, the authors argue that total signal is similar between preparations. In Fig. 4C, D, E, there is a rather large number of peptides that are of higher abundance using streptavidin beads, as also argued in Fig. S5. Please also note that Fig. S5 does not contain the information in the main body (all common peptides vs. 2 IFX peptides). Please amend and clarify.

Line 317: provide reference on how you estimated that your five donors were representative of the 50% of the European population. I personally would simply omit this note as this is not a key discussion point in the manuscript.

Line 322: Fig. 5D: please note that Fig. 5E and 5F are not referenced in the text.

Lines 322-324: Fig. 5D/E/F are essentially generated from the full ligandome from all 5 donors. Please explain why the donor clustering is not rather identical between the three conditions as the endogenous MHC-II peptide part vastly dominates the signals

compared to the compound-specific signal. Alternatively, I would combine the three figures in one and do the clustering on the total peptides found in all conditions.

Line 373: please provide the reference in which the same Fc domain construct elicited ADA response in few patients in clinical trials.

Line 394: streptavidin magnetic beads - be consistent in the manuscript!

Line 397: A slightly higher number of IFX peptide was recovered ... Fig. 4B - please define slightly and to what this increase was compared to.

Lines 413-424 and Fig. 8: In my opinion, this section does not belong in the discussion but could be incorporated into the section "MAPPs results on HLA-genotyped donors". Also, to strengthen the argumentation that the MHC-II peptides detected with this new workflow were bona fide associated to the MHC receptors of the 5 HLA-typed donors, please add to the manuscript a supplementary table of all IFX, KLH and Betv1a epitopes that were characterized in this study and provide the estimated binding affinity (or a similar measure) to the MHC receptors of the donors. Also, in response to the other reviewer, you can also include the identification criteria provided by Proteome Discoverer. Possibly this information was provided in Table S1, if that was so, apologies.

Fig 5 legends, line 46: Please note that MAPPs output provide MHC-II peptides, not T cell epitopes. Please amend.

Letter to Reviewers

We have some little shifts in line number stated by the Reviewer#2, but we hope to have successfully managed to implement all the suggested revisions.

Reviewer #2 (Comments to the Authors (Required)):

1) Line 46 , reference 13, 14: It would be more proper to quote the original document from the EMA and FDA. Also, please note that the FDA 2014 guideline is now accompanied by a draft guidance from 2022 - please add reference and include modifications as needed. **Dear reviewers, what do you mean for original documents? We have cited the two guidelines from the site of the respective health authority. Can you please point us the document you are talking about? We also added the draft guidance suggested by the reviewer of 2022 "Immunogenicity Information in Human Prescription Therapeutic Protein and Select Drug Product Labeling — Content and Format Guidance for Industry".**

2) Line 89 - "Compared to previous published MAPPs, we developed a robust method ...". Steiner et al. (reference 38 in the publication) already showed that MAPPs protocols are robust and reproducible. Please amend the statement.---**Dear reviewers, please see how this sentence has been rephrased in the introduction**

3) Line 98: high frequency - please define what is high frequency in this context. Reference 1 is outdated in this respect and there are more recent data to indicate that Infliximab is more on the moderate side regarding ADA incidence.--- **Dear reviewers, we updated the reference 1 with two more recent publications on Infliximab immunogenicity related papers. In ref 40 Cohen et al. reported the immunogenicity of Infliximab in a recent clinical trial and ref 41 is a summary of 20 years of Remicade from its launch on the market. We also changed "high" to moderate frequency.**

4) Line 155 (new line 365): Please indicate how mMoDCs were frozen as this is mentioned in line 171 of the manuscript--- **Dear reviewers, please see what we have added at line 154 (new line 365), "Cell pellets were frozen at -80°C for at least 16 hours before proceeding with the cell lysis step"...**

5) Line 191 (new line 402): The column is not self-packed as it bears a catalog number from ThermoFisher. Please correct.--- **Dear reviewers, please see our correction.**

6) Line 200 (new line 411): Survey scans m/y 266.7 to 4000 m/z: are you certain of the upper range? This would make the instrument very, very slow. Please check. --- **Dear reviewers, there was a typo here (corrected upper range is 2000 m/z). Many thanks for checking this out.**

7) Line 217 to 225 (and SI, (new line 432): please do not describe the description in detail the LFQ data package. It is more appropriate to put in the SI the graphical representation of the nodes as provided by Proteome Discoverer. If desired, you can add screenshots of each of the node. **Dear reviewers, it was done in the Supporting info as new Figure S1.**

8) Line 239 (new line 450): reference 42 is outdated, please add the 2017 reference mentioned on the website. --- **Dear reviewers, we added the other reference suggested from reviewer #2 found on website page, "GibbsCluster: unsupervised clustering and alignment of peptide sequences", Andreatta M, Alvarez B, Nielsen M, Nucleic Acids Research (2017) doi: 10.1093/nar/gkx248.**

9) Lines 257-259 / Lines 259-261 (new line 123): These two sentences appear to be redundant? --- **Dear reviewers, the second sentence is referring to the statistical evaluation of mMo-DC recovery between the**

two media. Could you please make a suggestion of how you would spell it out differently?

10) Lines 276 and following (new line 143): please number your donors in a consistent manner. At present, there are three different numerations, and this is confusing to the reader. --- Dear reviewers, we used the same prefix code (BC, which stands for buffy coat, followed by number of our cell bank depository). The number is not always progressing because the selected donors were chosen for their HLA-genotype. The number after the BC prefix is the one that we have internally as our bank was growing, but there is no issue in modifying the numbering if you think it would be better doing it in other way. What would be your proposal for donor classification and numbering?

11) Line 291 - 1.0-1.5x10⁶ DCs (new line 157): in which stage were this DCs counted? iMo-DCs, mMo-DCs, monocyte? Please precise.--- Dear reviewers, the count was referred to the mature dendritic cells. Please see how we implemented this in the manuscript.

12) Line 295 : Figure 3C is redundant to Figure S2, S3 and S4. In my opinion, Figure 3C is not absolutely required in the main body of the text. Dear reviewers, it was eliminated from the Figure 3 in the main manuscript as suggested.

13) Line 300 (new line 167): three different donors were tested, were these donors tested in several technical replicates? There can be no statistics out of n=1 sample. Please precise.--- Dear reviewers, the LFQ ratio was assessed on six different individual samples (2 conditions for BC65, 2 for BC115 and 2 for BC125, 2 treatment conditions which represent the two bead types). The LFQ ratio was calculated by inputting as categorical factor the type of beads used for the immunopurification. Then there are three biological replicates for each tested condition (e.g., Placebo=bead 1, Drug X= bead 2, please see scheme below). The different mice can be deemed as our donors (donor 1, donor2 and donor 3). We used pairwise ratio based for protein ratio calculation to avoid miscalculation due to the effect of missing values (this is concrete when using Protein abundance based protein ratio calculation in Proteome Discoverer). This hypothesis test method (called t-test background proteins in PD) uses the background population of ratios for all peptides and proteins (namely the housekeeping proteins) to determine whether any given single peptide or protein is significantly changing relative to that background. T-test (background based) assesses the distance of protein ratio from the ratio observed for the background proteins (highlighted in orange in the scatter plot). This method does not require that a given sample group contains technical replicates, but it does require that most of the protein abundances are unchanged between samples. This leads to less chance of false positive because the whole background of proteins is being considered. We are exactly in the perfect requirements suggested for applying t-test background-based hypothesis test when using pairwise ratio based calculation that is actually recommended by software developers for Label-Free Quantification studies. Please, see attached the publication from Pedro Navarro et al. that contributes to the statistic frameworks in PD:

“General Statistical Framework for Quantitative Proteomics by Stable Isotope Labeling”. by Navarro, P. et al, J. Proteome Res., 13, 1234-1247 (2014). [dx.doi.org/10.1021/pr4006958](https://doi.org/10.1021/pr4006958) that we added in the revised manuscript as ref 51 in line 429.

[Figure removed by editorial staff per authors' request]

From the background proteins the test ascertains the median change value for the ratios of background proteins of similar abundance and calculate a confidence interval. It follows that higher abundant proteins

will have a narrower confidence interval compared to lower abundant ones because the quantitative data are more reliable. As a consequence, a bigger ratio change is needed for lower abundant proteins to be considered significant. This is another way to see the distribution of the data and their significance in their fold change as we showed in Volcano plots in the main manuscript.

Then the p-values to statistically evaluate the change in expression between two samples are retrieved (shown as color code legend in the plot). Dark blue – not significant change; Dark red – most significant change.

14) Line 308 (new line 176): Figure 4A would benefit from a log₂ representation on the y axis rather than a log₁₀. Dear reviewers, please see the new box plot with log₂ transformation of protein abundances (protein abundances are calculated by summing sample abundances of the connected peptide groups).

15) Lines 308-314: There is an apparent contradiction here. In Fig. 4a, the authors argue that total signal is similar between preparations. In Fig. 4C, D, E, there is a rather large number of peptides that are of higher abundance using streptavidin beads, as also argued in Fig. S5. Please also note that Fig. S5 does not contain the information in the main body (all common peptides vs. 2 IFX peptides). Please amend and clarify. Dear reviewers, please see line 180 (Figure S5, already present in the original revised manuscript), but we added another consideration in the Discussion section line 277 of the reformatted manuscript (Introduction, Results, Discussion, Materials and Methods). As far as Figure 4C, D; and E is concerned, for instance, from the volcano plot for BC65 we have more than 1000 peptides that are higher in abundance, vs 508 peptides that are down-regulated vs FG (but if you see most of them fell in the zone of low proteins fold change, thus not contributing so much to the overall protein abundance of each sample). Firstly, Proteome Discoverer retrieved the abundance of a protein by summing its abundance-scored peptides. The majority of peptides still falls in the region between -1 and 1 as log₂ fold change (background proteins, with no significant change between samples). This leads to the fact that the total protein abundances (which Proteome Discoverer calculates as the sum of the different peptide-spectra matches (PSMs) of peptides coming from each and every protein) is more or less the same, and by normalizing for log scale this could even reduce the differences in protein abundance in the plot. If you also see the box plot, the values of 75% percentile are higher for streptavidin-processed samples. Instead, the median abundance (50% percentile) of total protein abundance is similar. As far as the two common IFX peptides is concerned, yes the XIC intensities were higher, but this was referred just to the pulsed “molecule of interest”, but the majority of peptides coming from the housekeeping proteins (or background proteins) has the same abundance as it could be reasonably figured out (also see normalized Peptide abundances in hierarchical clustering heatmaps in figure 5 from all the three different conditions, and also the scatter plot with color code p-

value in answer to comment 13).

16) Line 317 (new line 185): provide reference on how you estimated that your five donors were representative of the 50% of the European population. I personally would simply omit this note as this is not a key discussion point in the manuscript. Dear reviewers, we used www.allelefrequencies.net to calculate the allele frequencies. As suggested, we omitted the frequency % in the text because as suggested is not a key point for the discussion of results.

17) Line 322 (new line 191): Fig. 5D: please note that Fig. 5E and 5F are not referenced in the text. Dear reviewers, with the new figure 5 (single heatmap) suggested by reviewer #2 we just have now Figure 5D as heatmap that is cited in the reformatted manuscript at line 191.

18) Lines 322-324 (new line 196): Fig. 5D/E/F are essentially generated from the full ligandome from all 5 donors. Please explain why the donor clustering is not rather identical between the three conditions as the endogenous MHC-II peptide part vastly dominate the signals compared to the compound-specific signal. Alternatively, I would combine the three figures in one and do the clustering on the total peptides found in all conditions. Dear reviewers, if you see, the three heatmaps are exactly the same (for IFX and Bet v1a), but also KLH. Proteome Discoverer has just inverted the KLH heatmap compared to the other two heatmaps, but this was generated by default from the software. If you also see the peptide abundance is absolutely robust and conserved among the same donor treated with the three proteins (moreover, the analysis of the three proteins was conducted in three different days, this reinforces once more the robustness of the developed protocol and identification by MS). Plus, if you see Gibbs clustering on the three molecules, you can see that the clusters and binding motifs are pretty much conserved for the three tested proteins. However, we really appreciate your suggestion for integrating all the data in a single heatmap with hierarchical clustering and we implemented it in the new figure 5D. We absolutely find it more comprehensive for the reader at first sight. Please, see our implementation in the unique Figure 5D.

18) Line 373 (new line 253): please provide the reference in which the same Fc domain construct elicited ADA response in few patients in clinical trials. Dear reviewers, please take a look at reference 46 in the text (new line 253).

19) Line 394 (new line 275): streptavidin magnetic beads - be consistent in the manuscript! Dear reviewers, it was implemented in line 394 (new is 275) and following.

20) Line 397 (new line 277): A slightly higher number of IFX peptide was recovered ... Fig. 4B - please define slightly and to what this increase was compared to. Dear reviewers, please found our comments in the reformatted manuscript, line 277).

21) Lines 413-424 and Fig. 8 (new line 220): In my opinion, this section does not belong in the discussion but could be incorporated into the section "MAPPs results on HLA-genotyped donors". Also, to strengthen the argumentation that the MHC-II peptides detected with this new workflow were bona fide associated to the MHC receptors of the 5 HLA-typed donors, please add to the manuscript a supplementary table of all IFX, KLH and Betv1a epitopes that were characterized in this study and provide the estimated binding affinity (or a similar measure) to the MHC receptors of the donors. Also, in response to the other reviewer, you can also include the identification criteria provided by Proteome Discoverer. Possibly this information was provided in Table S1, if that was so, apologies. Dear reviewers, we moved this section as suggested by the reviewer #2. In table S1,S2 and S3 we have uploaded three different excel sheets with all identified MAPPs peptides. As far as estimated binding affinity is concerned, MAPPs peptides follows a normal distribution (with a length varying from 13 mer to 25 mer as visible in new Figure 7A of our data). As a consequence, IEDB software predicts the binding affinity of a peptide with a defined length (parameters to be input before running the analysis, default value for MHCII ligands is set to 15). Then this would require to run the analysis to all different peptide lengths, but we know from literature that it is not just a matter of

dissociation constant (K_D), but rather on how many peptide-MHC copies are exposed onto the dendritic cell membrane and on the stability of the complex itself. If you think that this would be an added value for the reader in the supporting material we can do the analysis on all different peptide lengths and see if they are predicted. However, I can anticipate there may be the case where peptides are not predicted, or even when they are predicted, there is a shift on the actual sequence of the predicted peptide (then the estimated K_D would be a predicted extrapolation and not the actual one) or the alleles of the tested donors did not match the predicted ones by IEDB for that peptide. Plus, the binding to the MHC is not the only factor taken into consideration in MAPPs assay, but also considering all the enzymatic machinery used in antigen-presenting cells for antigen processing. This can lead to the processing of peptide that have a moderate affinity for MHCII also, that could be more presented rather than high affinity MHCII binders that are poorly processed in endo-lysosomes. In fact, in our experience with difference discovery programs on biologics, there were instances of peptides revealed by MAPPs that did not match the IEDB predictions, both in terms of sequence and for the HLA-genotype of the tested donors. We truly believe that this gap can be filled only by introducing an increasing number of “true” peptides coming from MAPPs assays or cellular assays, that could build up even more the power of predictions (nowadays the vast majority of the data of training set used for MHCI and MHCII predictions are based on binding affinity data, but as you know this assays failed to consider all the intracellular machinery of the antigen processing and presentation). A summary of MAPPs epitope consensus sequences is shown in Table S1, S2, and S3 (.xlsx files).

22) Fig 5 legends, line 46: Please note that MAPPs output provide MHC-II peptides, not T cell epitopes. Please amend. Dear reviewers, please check it out in figure legends.

September 27, 2023

RE: Life Science Alliance Manuscript #LSA-2023-02095-TRR

Dr. Andrea Di Ianni

University of Turin

NBE-DMPK Innovative BioAnalytics, Merck Serono RBM S.p.A., an affiliate of Merck KGaA, Darmstadt, Germany, Via Ribes 1, 10010 Colletterto Giacosa (TO), Italy

Via Ribes 1

Colletterto Giacosa, Italy/Turin 10010

Italy

Dear Dr. Di Ianni,

Thank you for submitting your revised manuscript entitled "Assessing MAPPs assay as a tool to predict immunogenicity potential of protein therapeutics". We would be happy to publish your paper in Life Science Alliance pending final revisions necessary to meet our formatting guidelines.

- please upload all figure files as individual ones, including the supplementary figure files; all figure legends should only appear in the main manuscript file
- please add the Twitter handle of your host institute/organization as well as your own or/and one of the authors in our system
- titles in the system and on the manuscript file must match
- please use the [10 author names et al.] format in your references (i.e., limit the author names to the first 10)
- please remove this sentence from the manuscript file after the references section "Supplementary information is available at Journal's website."
- please add your main, supplementary figure, and table legends to the main manuscript text after the references section
- please add callouts for Figures 5E, F; S2; S3; S4; S6 to your main manuscript text
- please add scale bars to Figure 2A

A. FINAL FILES:

B. MANUSCRIPT ORGANIZATION AND FORMATTING:

Sincerely,

October 3, 2023

RE: Life Science Alliance Manuscript #LSA-2023-02095-TRRR

Dr. Andrea Di Ianni
University of Turin
NBE-DMPK Innovative BioAnalytics, Merck Serono RBM S.p.A., an affiliate of Merck KGaA, Darmstadt, Germany, Via Ribes 1,
10010 Colletterto Giacosa (TO), Italy
Via Ribes 1
Colletterto Giacosa, Italy/Turin 10010
Italy

Dear Dr. Di Ianni,

Thank you for submitting your Methods entitled "Assessing MAPPs assay as a tool to predict the immunogenicity potential of protein therapeutics". It is a pleasure to let you know that your manuscript is now accepted for publication in Life Science Alliance. Congratulations on this interesting work.

DISTRIBUTION OF MATERIALS:

Again, congratulations on a very nice paper. I hope you found the review process to be constructive and are pleased with how the manuscript was handled editorially. We look forward to future exciting submissions from your lab.

Sincerely,
